# The claustrum enhances neural variability by modulating the responsiveness of the prefrontal cortex

Huriye Atilgan ✉, Ivan P. Lazarte & Adam M. Packer

The claustrum is a densely interconnected structure involved in cognitive functions, but its influence on prefrontal circuits remains unclear. We conducted two-photon calcium imaging to assess mice dorsal prefrontal cortex (dPFC) cell responses during exposure to visual stimuli and widefield photostimulation of claustrum axons embedded in the dPFC. We identified three distinct subpopulations of neurons − sensory responsive, opto responsive, and opto-boosted cells−each exhibiting unique response dynamics to combined visual and optogenetic stimuli. Our findings reveal that claustrum stimulation increased single-cell variability while aligning average responses across neurons, thereby enhancing network homogeneity. During Pavlovian training, enhanced variability persisted, but homogeneity increased further, suggesting experience-dependent refinement. Finally, we performed claustrum axon silencing experiments which revealed that the claustrum may operate bidirectionally to maintain enhanced variability and homogeneity in the dPFC. These results highlight the crucial role of the claustrum in dynamically modulating dPFC activity, impacting both neuronal variability and network synchronization.

The claustrum and the prefrontal cortex (PFC) are tightly interconnected through a robust anatomical network[1–3], hinting at a pivotal role for the claustrum in modulating executive functions governed by the PFC[4–10]. While lesioning studies have not identified a single dysfunction associated with the claustrum, studies have unveiled intriguing insights into the claustrum's involvement in a wide range of tasks[11–15].

These broad functions are supported by the claustrum's extensive anatomical connectivity. It receives input from and projects to widespread cortical regions, including prefrontal, cingulate, temporal, and retrohippocampal areas[2], and shows modality-specific topography in its connections with sensory and association cortices[8,16–18]. This connectivity underlies its proposed role as a hub for coordinating cortical dynamics during cognitively demanding states. Recent studies have shown that the claustrum contributes to multimodal integration[16,19], reward association and decision-making[20,21] and the coordination of cortical slow-wave activity[22,23]. It also facilitates contextual memory[24–26]

and plays a critical role in performance under high cognitive load[27,28]. These findings converge on a view of the claustrum as a key regulator of cortical communication and internal state transitions.

Building on this emerging view of the claustrum as a regulator of cortical dynamics, recent studies have begun to dissect the mechanisms by which it modulates cortical activity. One proposal is that the claustrum contributes to cognitive control by synchronizing distributed cortical networks[29,30]. Supporting this idea, the claustrum activation has been observed across multiple behavioral tasks that require cognitive control, including attention allocation, engagement restriction and task switching[10,31,32]. Mechanistically, claustrum activation can bidirectionally modulate cortical excitability depending on brain region, cell type, and behavioral state. In frontal cortex, brief claustrum stimulation preferentially recruits fast-spiking interneurons, leading to a net inhibitory effect[9], whereas prolonged stimulation can result in complex excitation−inhibition patterns across the brain. Whole-brain mesoscale imaging further reveals that reducing

Department of Physiology, Anatomy, and Genetics, University of Oxford, Oxford, UK. ✉e-mail: huriye.atilgan@dpag.ox.ac.uk

claustrum activity enhances PFC responses and sensory-evoked activity, while increasing claustral output suppresses frontal responses and alters global functional connectivity[33]. These findings underscore the claustrum's capacity to shape neural dynamics both locally and across distant brain areas.

However, a key unresolved question is how the claustrum influences activity in the PFC: does it exert its effects directly via monosynaptic projections, or indirectly by engaging other cortical or subcortical regions that, in turn, modulate PFC function? This distinction is fundamental for understanding the organizational logic of claustro-cortical communication and the specificity of its cognitive impact. Given the claustrum's widespread connectivity, isolating its direct effects on PFC circuits requires a pathway-specific approach. In this study, we address this by selectively stimulating claustrum axons within the PFC, enabling us to probe the functional consequences of direct claustro-PFC input while minimizing polysynaptic recruitment of other brain regions. This approach allows us to directly investigate how claustral input modulates prefrontal activity. Supporting this strategy, recent anatomical and optogenetic evidence has demonstrated that a subset of claustrum axons projecting to the PFC exhibit robust light-evoked responses[16], providing a foundation for targeted, circuit-specific manipulation.

Building upon this foundation, our study aims to elucidate the claustrum's contribution to PFC neural responsiveness, exploring its influence in both naive and trained states. To probe the functional consequences of this pathway, we combined local claustrum axon photostimulation embedded within the PFC with passive visual stimulation while recording PFC population activity using two-photon calcium imaging. We leveraged the known sensory responsiveness of dPFC neurons[34,35] and used passive visual stimuli to avoid confounding effects of task demands, motivation, or motor output—factors known to interact with both claustrum and PFC circuits[32]. This approach enabled a controlled examination of how claustral input shapes local PFC computation.

Our results demonstrate that claustrum input dynamically regulates dPFC responsiveness to sensory cues, acting as a bidirectional modulator that can either enhance or suppress sensory processing. This modulation serves to enhance neural variability and promote network homogeneity. The magnitude and direction of this effect depend on the baseline excitability of individual neurons and are shaped by prior experience, suggesting a learning-dependent gating of claustral influence. These findings highlight the claustrum's role in fine-tuning sensory processing and cognitive function, enabling the PFC to adaptively respond to internal states and external demands.

## Results

### Claustrum axon photostimulation alters dPFC cell responsiveness

We employed an optogenetic experimental framework to elucidate how claustrum axons modulate responses in the dorsal prefrontal cortex (dPFC), (Fig. 1A). We monitored neural activity in dPFC in mice during exposure to visual stimuli, while photostimulating claustrum axons in the same area. Genetically modified mice expressed the calcium indicator GCaMP6s in CaMKII+ cells and the red-shifted opsin ChrimsonR[36] in claustrum axons (Fig. 1B, C, see methods for our injection strategy, Supplementary Fig. 1). We employed two-photon microscopy in head-fixed mice for population calcium imaging. The mice were exposed to visual stimuli ("visual"), photostimulation ("opto"), and a combination of both visual stimuli and photostimulation ("visual+opto") in various inter-trial intervals (Fig. 1D; 30 trials each, randomly presented, visual stimuli: 590 nm, 56 lm high-brightness LED, total duration 250 ms; photostimulation: 595 nm, 10 × 25 ms pulses @ 40 Hz, at 1.5 mW, total duration 250 ms). To prevent unintended retinal activation from stray photostimulation light, we incorporated a custom-designed light blocker around the

objective. When the light blocker was in place, optogenetic stimulation did not induce pupil constriction, confirming that visual pathways were not inadvertently engaged (Supplementary Fig. 2). This setup facilitated the observation of neural dynamics at the single-cell resolution across a large field of view (580 × 580 μm) as we administered wide-field photostimulation (cranial window: ~4000 μm diameter) to claustrum axons, thereby assessing the influence of claustrum axon photostimulation on dPFC cellular responses to visual stimuli.

Calcium responses revealed diverse changes in dPFC neurons following claustrum axon photostimulation. Some neurons exhibited enhanced responses to visual+opto stimuli (Fig. 1E), while others showed reduced responses under the same conditions (Fig. 1F). Notably, these effects were observed in both excited and inhibited responses (Supplementary Fig. 3A, B), highlighting the high diversity of the claustrum axons' impact on dPFC neuronal activity to visual stimuli.

A subset of neurons exhibited activity exclusively in response to opto stimuli, showing no discernible reaction to visual stimuli alone or when presented in combination with photostimulation (Fig. 1G). On the other hand, a distinct subset of cells demonstrated an interesting phenomenon where photostimulation effectively enhanced their responsiveness to visual stimuli, even though these cells did not exhibit any response when presented with visual stimuli or photostimulation alone (Fig. 1H, and Supplementary Fig. 3C). These findings show the complex and precise modulation of neuronal responses in the dPFC by claustrum axon photostimulation, highlighting its role in selectively enhancing or suppressing neural activity under varying condition.

Upon analyzing all the recorded cells, we found that 49% of the total recorded cells exhibited responsiveness (Fig. 1I, $n = 24,575$ from 49,212 recorded cells, $n = 13$ mice, 270 field of view) Responsiveness defined by an independent t-test comparing activity 1 sec before and after stimulus offset (−1 to 0 s vs. stimulus duration to stimulus duration +1 s) for visual, opto, or visual+opto trials, with false discovery rate correction applied for multiple comparisons (α = 0.05). Within this responsive cohort, 21% showed responsiveness to visual stimuli ($n = 10,531$) and were classified as *sensory responsive cells*, with 42% of these cells being exclusively responsive to visual stimuli. 29% responded to opto stimuli ($n = 11,365$) and were classified as *opto responsive cells*, among which 44% were exclusively responsive to opto stimuli. 3% of these cells ($n = 1802$) exhibited responsiveness to either visual or opto stimuli alone but not to visual+opto stimuli. Furthermore, 14% of the cells ($n = 5783$), were classified as *opto-boosted cells*, and displayed responsiveness to visual stimuli only when presented with photostimulation, indicating a possible interaction between the visual stimuli and photostimulation that is not evident when each stimulus is presented alone.

To provide a complete picture of population-level response types, we also identified several overlapping categories—such as cells responsive to both visual and opto stimuli, or to all three conditions (visual, opto, and visual+opto). While each of these combinations represented fewer than 8% of cells, they are reported in Supplementary Table 1 for transparency. However, we emphasize that the interaction among these categories is complex and not always easily interpretable —for example, cells responsive to visual and opto stimuli independently may reflect convergence or unrelated excitability, without clear synergy. For this reason, we focused our main analyses on the three most dominant and mechanistically informative groups: sensory-responsive, opto-responsive, and opto-boosted neurons.

To better understand how these combined subpopulations within the same network process visual and opto stimuli, we examined the balance between excited (E) and inhibited (I) responses, which is crucial for neural network information processing[37,38]. The E:I ratio refers to the proportion of cells that exhibited a significant increase (excitation) versus a significant decrease (inhibition) in ΔF/F response

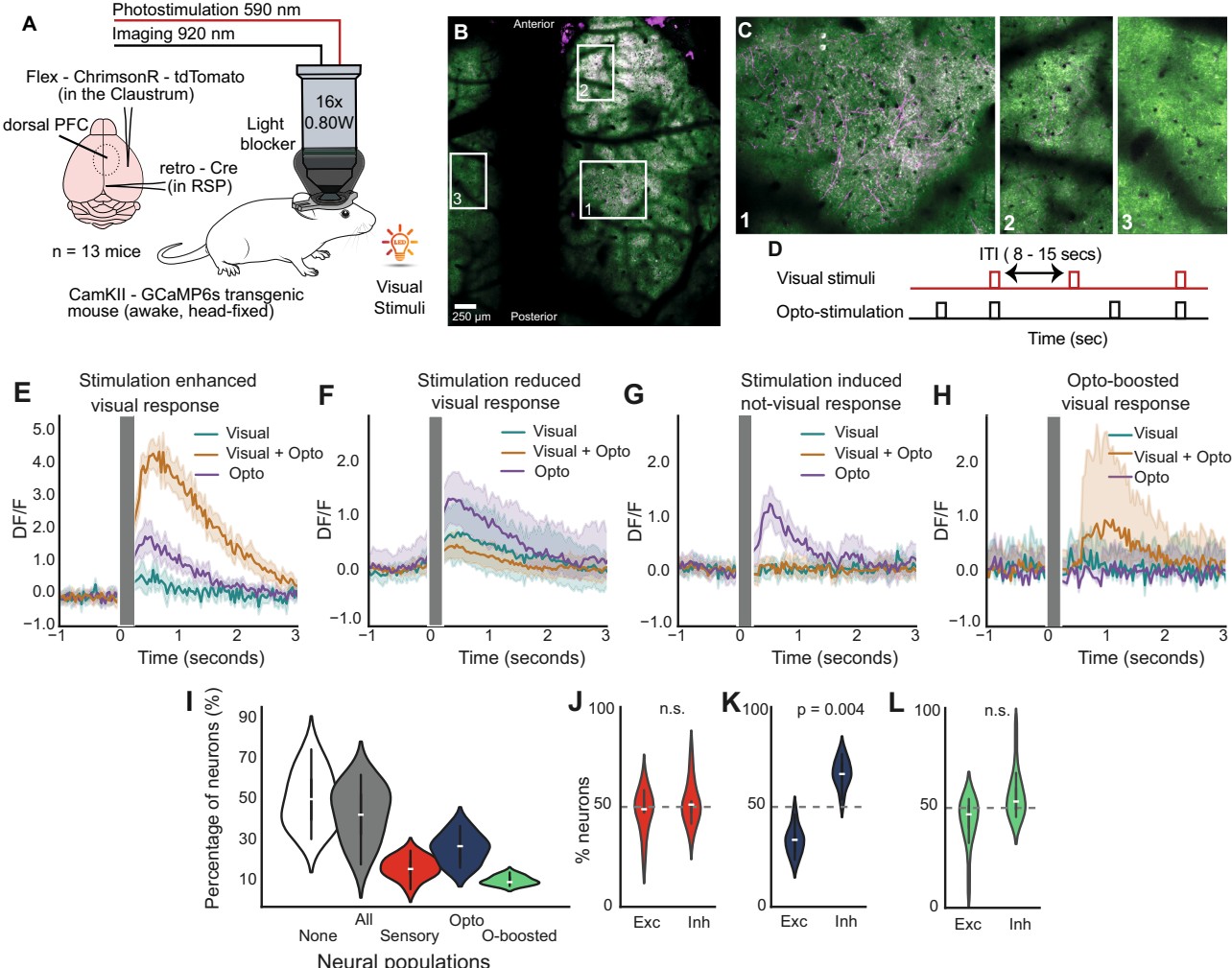

**Fig. 1 | Claustrum axon photostimulation modulates dPFC cell responses.**
**A** Schematic of the experimental setup for two-photon imaging of dorsal PFC neurons expressing GCaMP6s in head-fixed, awake mice, with concurrent photostimulation of claustrum axons during visual cue presentation. **B** Stitched two-photon image of the dorsal surface of the brain showing GCaMP-expressing neurons (green) and claustrum axons labelled with tdTomato (magenta) within the imaging field of view. **C** Zoomed-in images from Panel B. Image 1 shows the embedded axons; image 2 shows a more anterior field of view; image 3 shows a contralateral field of view without any axon labelling as an outcome of the ipsi-hemisphere specific projections of the claustrum. **D** Timing sequence of optogenetic stimulation and visual stimuli presentation across trials with variable inter-trial intervals (ITIs). **E−H** Time course of neuronal calcium responses (DF/F) illustrating the modulatory effects of claustrum axon photostimulation on dPFC sample neuron responses to visual stimuli. **I** Percentage of dPFC neurons per animal assigned to each response-defined group: All (any responsive), None (non-responsive), Sensory (respond to visual stimulus alone), Opto (respond to optogenetic stimulation alone), and Opto-boosted (respond only to paired visual +optogenetic stimuli). Violin plots show the across-animal distribution ($n = 13$ mice; each dot = one animal; center line = median; box = IQR). **J** Across mice, proportion of excitatory (Exc) vs inhibitory (Inh) neurons within sensory-responsive cells. Violin plots as in I ($n = 13$; Wilcoxon signed-rank; *n.s.* not significant). **K** Same as J for opto-responsive cells. **L** Same as J for opto-boosted cells.

following stimulus presentation. This analysis revealed that opto-responsive cells uniquely exhibited a higher prevalence of inhibited responses, with an E:I ratio of 33:67 (0.49, $p = 0.004$, $n = 13$ mice, Fig. 1L). When considering only opto responsive cells that were exclusively responsive to optogenetic stimulation (excluding any cells responsive to visual or visual+opto stimuli), the E: I ratio became even more distinct 28:72 (0.39).

For comparison, sensory-responsive cells exhibited an E:I ratio of 48:52 (0.92, Fig. 1K), and opto-boosted cells showed an E:I ratio of 44:56 (0.79, Fig. 1M). Notably, the trend toward a higher proportion of inhibited responses was unique to the opto-only group and was not observed in the sensory-responsive or opto-boosted populations. The elevated inhibition in opto-responsive cells aligns with prior findings[6] and supports the interpretation that optogenetic activation of claustral axons can drive inhibitory dynamics within the dPFC network−particularly in cells not otherwise responsive to sensory input. In

contrast, the lack of a similar inhibitory bias in the sensory or opto-boosted groups may reflect either more subtle effects of inhibition or the dominance of excitatory drive from visual stimuli. These results suggest that inhibition is a context-dependent component of the broader modulatory influence of claustrum input within dPFC circuits.

## The claustrum modulates dPFC neuronal responses to enhance neural flexibility

To clarify the role of three identified subpopulations in the dPFC during claustrum axon photostimulation, we performed targeted analyses focusing on their individual response patterns to better understand their functional dynamics in sensory processing.

Initially, we focused on sensory responsive cells and, consistent with the literature, observed varying visual cues responses among dPFC cells[34,35,39], displaying different levels of excitation and inhibition (Fig. 2A). Upon visual+opto stimulation (Fig. 2B), these cell responses

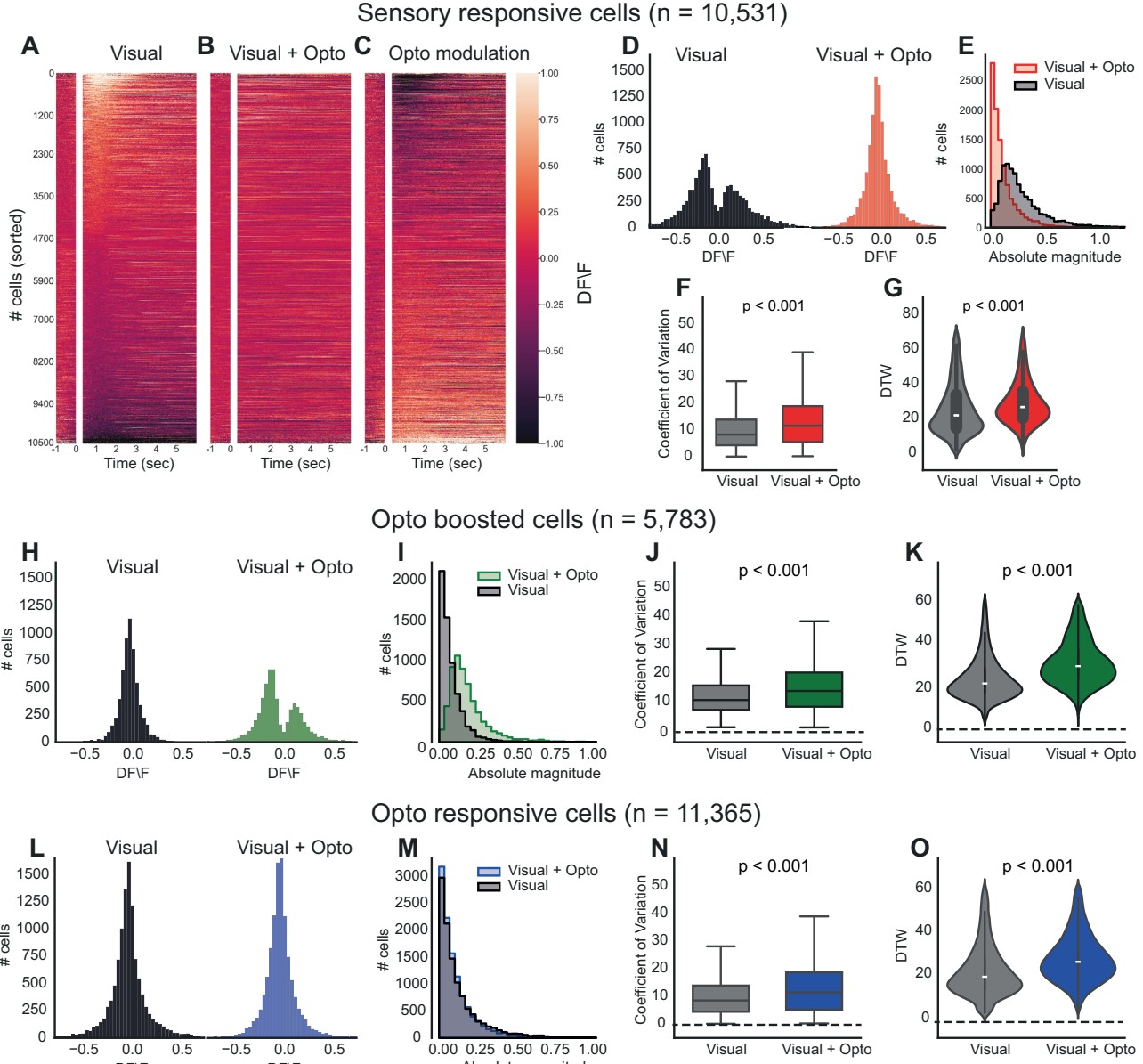

**Fig. 2 | Claustrum axon modulation enhances neural flexibility in the dPFC.**
**A–C** Heatmap of neuronal activity over time for neurons under three different conditions: Visual (**A**), Visual + Opto (**B**) and their differences (**C**, opto modulation) (**D**) Distribution of the DF/F for neurons under the conditions of visual and visual + opto for sensory responsive cells. **E** Histograms representing the absolute magnitude of neuronal responses to visual and visual + opto stimuli for sensory responsive cells (Wilcoxon, *p* = 2.10e-236). **F** Box plots comparing the coefficient of variation for neurons under visual and visual + opto conditions for sensory

responsive cells Wilcoxon signed-rank test (Wilcoxon, *p* = 2.48e-148). **G** Violin plots illustrating the distribution of dynamic time warping (DTW) values under visual-only and visual + opto conditions (Wilcoxon, *p* = 1.35e-69). **H–K** Similar to (**D–G**), but for opto-boosted cells (Wilcoxon, *p* = 0.192; 2.33e-131; 1.56e-6). **L–O** Similar to (**D–G**), but for opto responsive cells (Wilcoxon, *p* = 8.66e-244; 2.19e-73; 2.23e-23). (*n* = 13 mice; one dot = one mouse; cells and trials are nested within mouse and were first summarised within mouse; two-sided Wilcoxon signed-rank on per-mouse summaries).

notably changed (Fig. 2C), with excited responses in cells that were originally inhibited by visual stimuli alone and inhibited responses in those that were excited. To quantify this effect, we calculated the mean ΔF/F for 1500 ms after stimulation. The resulting histogram for visual +opto stimuli showed a significantly different distribution, where photostimulation shifted the bimodal distribution toward a Gaussian-like curve, indicating a normalising effect on neuronal responsiveness (Fig. 2D). At the population level, the absolute magnitude of neuronal responses to visual stimuli significantly decreased in the presence of photostimulation (Fig. 2E, *p* < 0.001, Kolmogorov-Smirnov test). To better understand how the distribution of neuronal responses was affected, we performed an empirical cumulative distribution function

analysis, which revealed that a greater number of neurons were responsive when visual+opto stimuli were presented compared to visual stimuli alone (Supplementary Fig. 4A). This pattern suggests that, although the amplitude of individual responses decreased, the optogenetic intervention recruited a larger population of neurons, thereby enhancing distributed sensory processing with a balanced level of responsiveness.

To explore the mechanisms underlying this distributed processing, we hypothesized that the claustrum might increase neural flexibility in the dPFC by elevating the variability in neuronal responses across repeated trials. We initially quantified this variability by examining the coefficient of variation, which showed a significant increase

under visual+opto stimulation (Fig. 2F, $p < 0.001$, paired t-test). We then applied dynamic time warping (DTW) analysis—a technique that measures the similarity between two temporal signals—across trials for both visual and visual+opto stimuli. A higher DTW value indicates a greater disparity between two sequences, necessitating more adjustments to achieve alignment. Significantly larger DTW values for the visual+opto condition (Fig. 2G, $p < 0.001$, paired t-test) demonstrated increased variability when visual cues were combined with optostimulation, suggesting that the neurons are adapting their firing patterns more distinctly in each trial. This finding supports our hypothesis that claustral modulation enhances neural flexibility by increasing the variability of neuronal responses within the dPFC.

Subsequent analysis focused on the opto-boosted subpopulation, a unique subpopulation that did not respond to visual or optogenetic stimulation alone but showed activity when these stimuli were combined. The response distribution of these neurons shifted from a non-responsive state to a bimodal pattern of activation upon visual-optogenetic stimulation, as illustrated by histograms (Fig. 2H–K). Notably, there was a significant increase in the absolute magnitude of responses to visual-opto stimuli compared to visual stimuli alone (Fig. 2I, $p < 0.001$, Kolmogorov-Smirnov test). This suggests that photostimulation can unveil latent responsiveness in certain dPFC neurons, by making the neurons more excitable. This subpopulation also exhibited a higher coefficient of variation under visual+opto stimuli (Fig. 2J, $p < 0.001$, paired t-test) and more pronounced temporal variability in dynamic time warping analyses (Fig. 2K, $p < 0.001$, paired t-test) highlighting an enhanced flexibility in their response patterns when visual stimuli presented with optostimulation.

Although opto-responsive neurons do not show changes in the absolute magnitude of their response to visual cues (Fig. 2M, $p = 0.192$, paired t-test), they also exhibited increased variance in their neural response for visual+opto stimuli (Fig. 2N, $p < 0.001$, paired t-test) and DTW analysis (Fig. 2O, $p < 0.001$, paired t-test).

Despite the differing results observed in the first two subpopulations − where sensory-responsive cells exhibit normalized responses and opto-boosted cells reveal latent responsiveness—these outcomes collectively suggest that claustrum stimulation in the dPFC enhances the flexibility of neural responses. The observed increase in the coefficient of variation and pronounced temporal variability indicates greater variability in neuronal responses across all subpopulations when subjected to combined visual and optogenetic stimuli. Collectively, these findings suggest that the claustrum plays a crucial role in facilitating flexible and distributed processing mechanisms within the dPFC.

## Greater dPFC network homogeneity with claustrum axon modulation

The greater response variation we observed in these subpopulations was consistent across all animals as well as in non-responsive cells (Fig. 3A-C), suggesting that optogenetic stimulation broadly affects the neural network beyond just the most noticeable responsive cells. This implies a widespread influence of the claustrum on the dPFC, potentially affecting even those neurons that do not typically react strongly to visual and/or optogenetic stimuli alone.

To assess the claustrum modulation on the dPFC neural network, we first examined the opto modulation compared to the neural response to visual cues. Correlation analysis revealed a strong negative correlation. As the ΔF/F response to visual stimuli increased, photostimulation had a more pronounced inhibitory effect. In contrast, when the ΔF/F response was negative, photostimulation led to an excitatory effect across all subpopulations (sensory: Fig. 3D, $r = -0.876$, $p < 0.001$; opto: Fig. 3E, $r = -0.851$, $p < 0.001$; opto-boosted: Fig. 3F, $r = -0.735$, $p < 0.001$). Hence, claustrum modulation in dPFC cells strongly depends on the baseline state of responsiveness of the given cell, and this dependency is not limited to excitatory cells.

Additional experiment revealed that inhibitory cells exhibit a similar pattern of modulation, indicating that this effect was consistent across different cell types (Supplementary Fig. 5, $n = 5$ Nkx2.1- GCaMP6s transgenic mice). Furthermore, activating thalamic axons in the same region did not reproduce the variability or opto-boosted effects observed with claustrum stimulation, suggesting that the modulation was not a general feature of all subcortical inputs (Supplementary Fig. 6, $n = 3$ CamKII-GCaMP6s transgenic mice).

Then, we assessed the collective impact of all responsive cells including the three identified subpopulations by examining the cross-correlation of trial-averaged responses across all neurons. To capture global changes in representational structure, we first computed a representational similarity analysis (RSA) across all recorded neurons from all animals (Fig. 3G). The matrix for visual stimuli showed distinct cell group clustering, suggesting a highly coordinated neural network (Fig. 3G, left). Conversely, the matrix for visual + opto stimuli demonstrated a more uniform distribution, with an absence of distinct clustering (Fig. 3G, right, using the same clustering as defined by the visual condition), indicative of a more homogenized representational landscape.

To further examine, we calculated within-FOV cross-correlation of trial-averaged responses for 1.5 sec after the stimulation window to quantify local network synchrony. This analysis revealed that optogenetic stimulation increased the average pairwise correlation within each FOV (Fig. 3I), consistent with a synchronization of activity across the recorded population.

When we calculated the average change in correlation due to optostimulation across animals, both the global RSA (which summarizes representational similarity across all recorded neurons) and within-FOV cross-correlation showed a consistent increase for all responsive as well as non-responsive cells (Fig. 3H, significantly higher in all responsive cells than non-responsive cells, $p < 0.001$).

Notably, while the coefficient of variation analysis indicated higher variability in responses across trials within individual cells, the cross-correlation analysis, which examines trial-averaged responses, suggests increased synchrony among dPFC neurons during visual + opto trials. This indicates that the visual + opto stimuli facilitated a more homogenized and correlated interaction among all cells. This synergy among subpopulations likely contributes to refining dPFC activity, strengthening functional connections between neurons, and integrating them into a more cohesive network.

## Claustral modulation-induced network homogeneity is enhanced by experience

To investigate how claustral modulation affects neural flexibility driven by experience, we trained mice ($n = 8$) in a Pavlovian conditioning task with the same visual stimuli. Mice were water-restricted and presented with the same visual cues as in the experiments above followed by a drop of water (80% of the trials). 7 mice showed anticipatory licking activity before the reward within 10 days (Fig. 4B–D, $p = 0.028$), indicating that they had learned to associate the stimuli with the reward. By first establishing this association, we ensured that the visual stimuli acquired behavioral relevance, thereby engaging cortical circuits of interest such as the claustrum and PFC, even during subsequent passive exposure[40,41]. This approach allowed us to probe neural representations of behaviorally meaningful stimuli without the confounds of motor output or ongoing task performance. After mice learned the association, they underwent the previous recording protocol in which they were passively exposed to visual cues with and without photostimulation. Notably, in this recording, the animals were familiar with visual stimuli although it was passively presented.

Similar to the naive animal recordings, we found three subpopulations in similar ratios for the trained animal recordings (53% responsive cells, 22% sensory responsive cells, 30% opto boosted cell

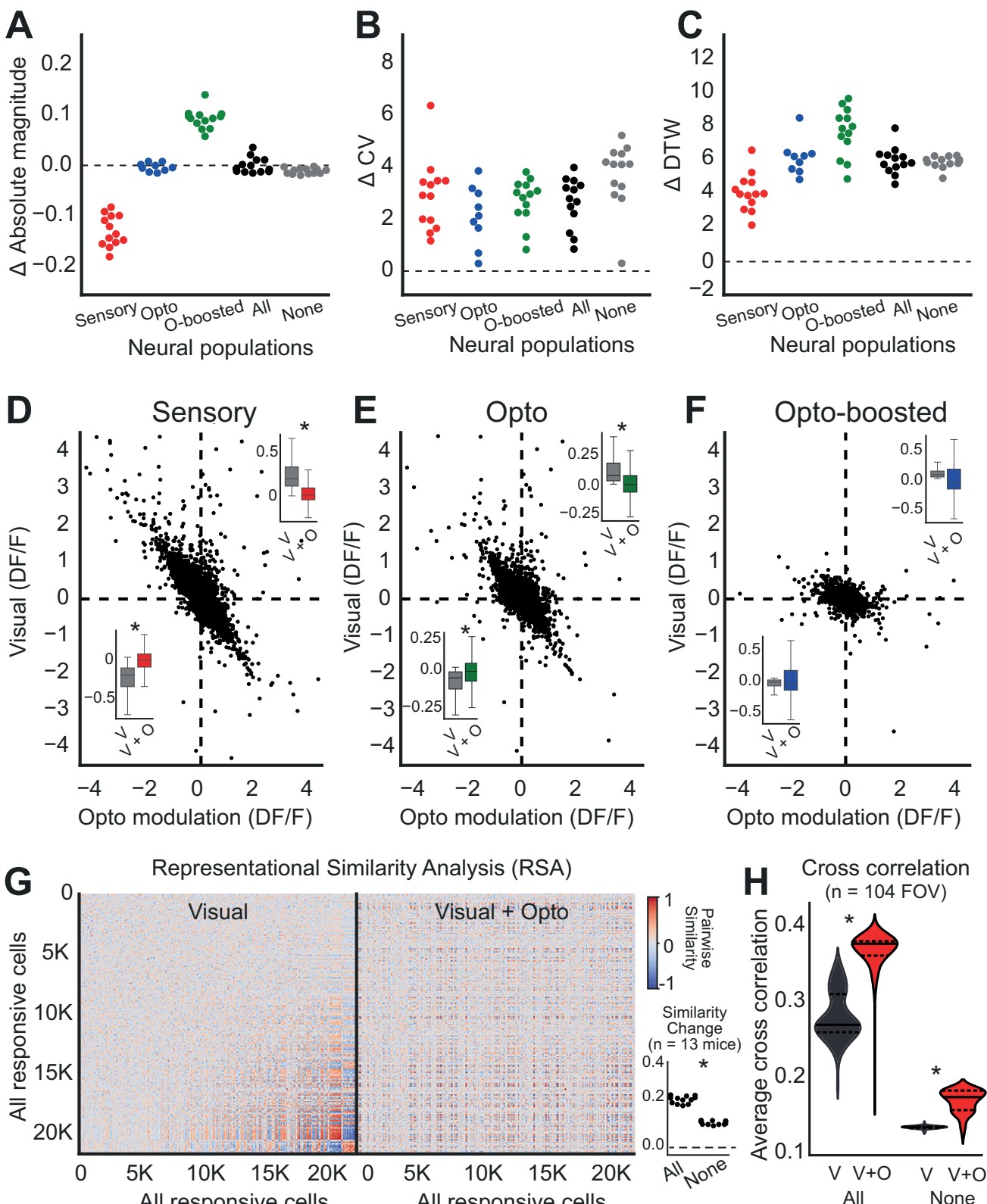

and 12% opto-boosted cells, Fig. 4E). However, we observed no significant differences in excitatory and inhibitory responses among opto responsive cells, unlike the findings from the naive dataset (Fig. 4F, sensory: $p = 0.578$; opto $p = 0.110$; opto-boosted $p = 0.297$, pairwise t-test). These three subpopulations displayed similar patterns post-training;(i) sensory-responsive neurons exhibited reduced response magnitude with an increased disparity in neural response upon

claustrum axon photostimulation (Fig. 4G), (ii) opto-boosted neurons showed amplified response magnitudes alongside greater neural variability (Fig. 4H), and (iii) optogenetically responsive neurons displayed consistent absolute magnitudes but demonstrated enhanced variance in neural responses (Fig. 4I).

The fundamental mechanisms across subpopulations remained consistent; the change in absolute magnitude (Fig. 4J, sensory $p = 0.031$;

**Fig. 3 | Modulation of claustrum axons enhances cross-correlation in the dPFC neural network. A** Swarm plot for the absolute magnitude of response changes for sensory, opto, opto-boosted, all (includes any responsive cells) and none (cells do not respond to any stimulation) neuronal populations. Each dot represents an individual animal (*n* = 13). **B** Swarm plot for changes in the coefficient of variation (ΔCV) for each neuronal population. **C** Swarm plot showing dynamic time warping (ΔDTW) differences across neuronal populations. **D–F** Scatter plots showing the relationship between visual responses (ΔF/F) and opto-modulation (ΔF/F) for sensory (**D**), opto (**E**), and opto-boosted (**F**) neuronal subpopulations. Dotted lines indicate zero levels. Insets show average ΔF/F changes for excited (top) and inhibited (bottom) responses across animals (*n* = 13; box plots show median and interquartile range; Wilcoxon signed-rank test (*p* < 0.05)). **G** Heatmaps display cross-correlation coefficients among all responsive cells during visual (left) and visual+opto (right) conditions. **H** The average correlation changes after optostimulation, highlighting shifts in neural connectivity and synchronization under combined stimulation (Pairwise t-test; 104 FOV, All: *p* = 4.43e-5; None: *p* = 3.51e-5).

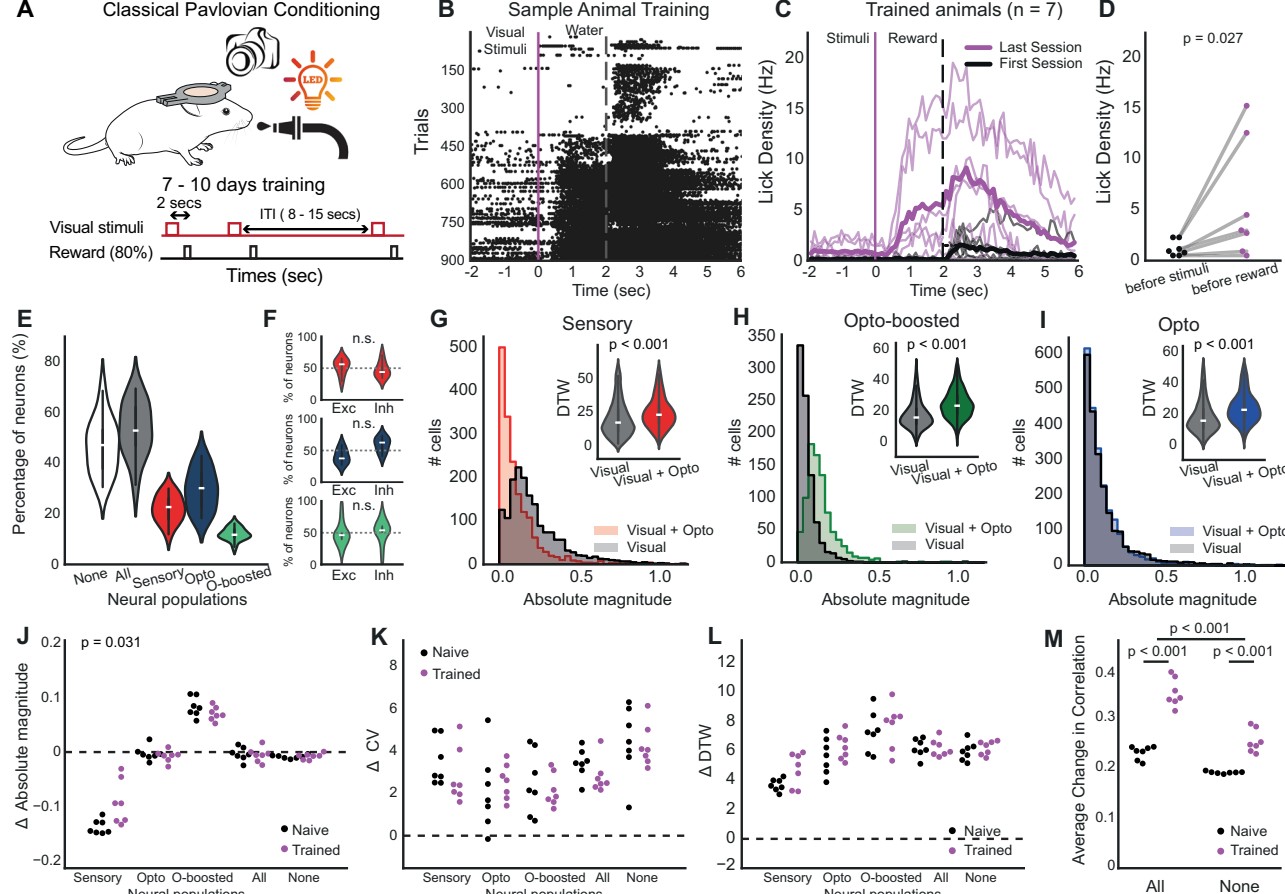

**Fig. 4 | Claustral modulation-induced neural homogeneity is enhanced by training. A** Schematic illustrating the experimental design for classical Pavlovian conditioning in mice, involving visual stimuli over a 7–8 day training period with an 80% reward rate. **B** A raster plot displaying lick responses from a sample animal during training, aligned to visual stimulus onset and reward delivery cues. **C** Line plots presenting the average lick rate per second for trained animals (*n* = 7) across the first (black) and the last (magenta) session of the training. Thick line is for average, thin line is for individual mice. **D** Average the lick density (1 sec) before stimuli and before reward. **E** Percentage of dPFC neurons per animal assigned to each response-defined group: All (any responsive), None (non-responsive), Sensory (respond to visual stimulus alone), Opto (respond to optogenetic stimulation alone), and Opto-boosted (respond only to paired visual+optogenetic stimuli). Violin plots show the across-animal distribution (*n* = 7 mice; each dot = one animal; center line = median; box = IQR). **F** Violin plots comparing the percentage of excitation (Exc) and inhibition (Inh) responses among different neural populations per animal (Violin plots as in **E**, Wilcoxon, *n* = 7). **G–I** Absolute magnitude and dynamic time warping (DTW) analysis comparing sensory (**G**, Wilcoxon, *p* = 2.61e-49), opto-boosted (**H**), *p* = 3.15e-134), and opto (**I**), *p* = 1.612e-91) neural populations under visual only and visual+opto condition. **J** Swarm plot for the differences of absolute magnitude of neuronal responses to visual and visual+opto stim across neural populations in naive versus trained. (Wilcoxon sign-rank, each dots represent one animal). **K** Same with J but for coefficient of variation (ΔCV). **L** Same with J but for ΔDTW. M) The average change in cross-correlation coefficients between response patterns of naive versus trained animals (Two-way ANOVA; *p* = 1.52e-10; 2.98e-08;1.48e-03).

opto *p* = 0.110; opto-boosted *p* = 0.297, paired t-test), the change in the coefficient of variation (Fig. 4K, sensory *p* = 0.219; opto *p* = 0.375; opto-boosted *p* = 0.578, paired t-test), and the change in DTW (Fig. 4L, sensory *p* = 0.0782; opto *p* = 0.469; opto-boosted *p* = 0.812, paired t-test) was greater than zero but did not differ in trained animals compared to naive animals across populations. This suggests that while training influences how visual cue perceived and behavioral responses, the intrinsic variability in neural responses induced by photostimulation remains relatively stable. Hence, the modulatory effects of the claustrum on these neurons are robust against experiential changes—at least for the aspects of neural response variability measured here.

On the other hand, correlation analysis revealed an average increase in correlation among neuronal responses in trained animals, indicating a more synchronized and integrated network response following training (Fig. 4M). A two-way ANOVA showed significant

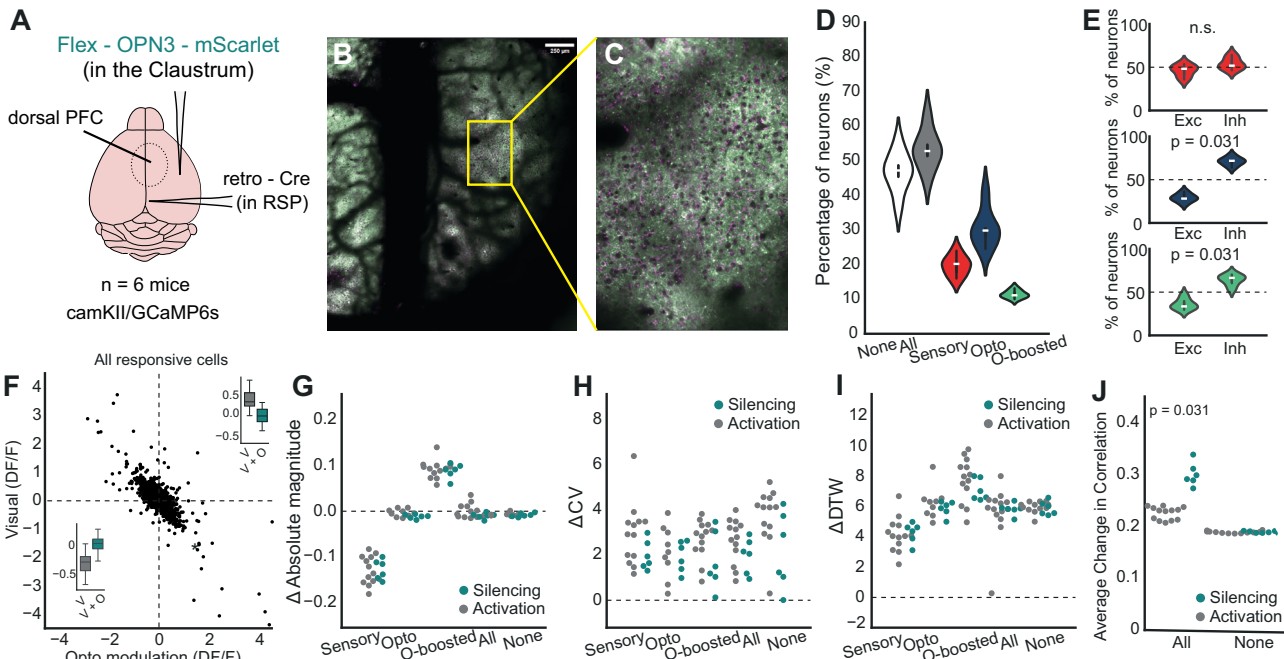

**Fig. 5 | Claustrum axon silencing enhances network synchrony in the dPFC to a greater degree than activation. A** Schematic of the experimental setup illustrating the targeting of Flex-eOPN3-mScarlet expression in the claustrum and retro-Cre in the retrosplenial cortex (RSP) and cranial window over dPFC (n = 7 camKII/GCaMP6s mice) (**B**) Representative confocal microscopy images showing axonal expression of Chrimson (red) in dPFC. **C** Additional zoom confocal images showing the targeted axons within the dPFC. **D** Percentage of dPFC neurons per animal assigned to each response-defined group: All (any responsive), None (non-responsive), Sensory (respond to visual stimulus alone), Opto (respond to optogenetic stimulation alone), and Opto-boosted (respond only to paired visual +optogenetic stimuli). Violin plots show the across-animal distribution (n = 6 mice; each dot = one animal; center line = median; box = IQR). **E** Box plots comparing the percentages of excitatory (Exc) and inhibitory (Inh) responses within each neural population (Wilcoxon). **F** Scatter plot for the relationship between visual response (dF/F) and opto modulation (dF/F) across all responsive neurons. **G** Swarm plots for the absolute magnitude of neuronal responses across different groups, differentiated by ChrimsonR (activation) and eOPN3 (silencing) expression. **H** Same with (**H**) but for coefficient of variation (ΔCV). **I** Same with (**G**) but for dynamic time warping values (ΔDTW) (**J**) The average change in cross-correlation coefficients between two groups (Wilcoxon sign-rank).

main effects of training (F(1,24) = 112.57, p < 0.001) and responsiveness (F(1,24) = 64.40, p < 0.001). Additionally, there was a significant interaction between training and responsiveness (F(1,24) = 12.86, p = 0.002), indicating that the effects of training varied depending on cell responsiveness. This implies that experience amplifies the effects of claustrum axon stimulation, leading to more coordinated neural activity, particularly in responsive cells, while preserving the underlying mechanisms.

## Silencing the claustrum also enhances neural flexibility in the dPFC

To further elucidate the functional role of the claustrum, we conducted an additional photostimulation experiment to explore the effects of claustrum axon silencing on dPFC responses in both naive and trained animals. The same methodology was used as in the activation experiment, except we injected eOPN3 to silence axonal neural activity (Fig. 5A–D). Notably, eOPN3 is a light-sensitive opsin that functions as an inhibitory tool by reducing neuronal activity upon photostimulation. This contrasts with ChrimsonR, which we use for activation experiments, as ChrimsonR is a red-light-sensitive opsin that depolarizes neurons to induce excitation. Thus, while Chrimson activates neural circuits via depolarization, eOPN3 effectively silences them through hyperpolarization (activates a signaling cascade leading to hyperpolarization and subsequent silencing of neural axonal activity, providing an effective means of optogenetic inhibition[42], see methods for more details).

Initially, we assessed the percentage of responsive neurons (Fig. 5E). The distribution of responsive neurons closely mirrored that observed during claustrum axon activation: 53% of the cells were responsive, with 20% classified as sensory-responsive, 30% as opto-

responsive, and 11% as opto-boosted. Notably, in the opto-boosted subpopulation of naive animals (addition to opto responsive cells, p = 0.031), inhibited responses were more pronounced than excited responses, a difference not seen during claustrum axon activation (Fig. 5F, green plot for opto-boosted, p = 0.031). Moreover, the degree of the modulation effect was strongly correlated with the cell responsiveness (Fig. 5F, r = -0.999, p < 0.001)

Then, we explored each subpopulation dynamics and found similar dynamics to the activation experiment (Fig. 5G, absolutemagnitude, sensory: p = 0.966; opto p = 0.066; opto-boosted p = 0.765; Fig. 5H ΔCV, sensory: p = 0.106; opto p = 0.776; opto-boosted p = 0.179; Fig. 5I, ΔDTW, sensory: p = 0.701; opto p = 0.456; opto-boosted p = 0.244). The only difference was a higher change in the cross-correlation for the silencing group than the activation group Fig. 5J, A two-way ANOVA, main effects of training: F(1,34) = 89.521, p < 0.001, responsiveness: (F(1,34) = 264.120, p < 0.001,interaction F(1,34) = 87.529, p = 0.002), suggesting that silencing the claustrum enhances network synchrony in the dPFC to a greater degree than activation, though through similar mechanisms.

We then these mice on the same Pavlovian task and showed no differences between naive and trained animals in terms of the measured neural dynamics (Supplementary Fig. 7). These findings suggest that the claustrum exerts a dual regulatory influence on the dPFC, modulating its output in a flexible manner depending on whether the claustrum is activated or silenced.

Overall, these findings suggest that the claustrum serves as a critical modulator of neural activity in the dPFC, influencing both the amplitude and variability of neuronal responses as well as the overall synchrony of the neural network. The effects observed during claustral silencing versus activation provide insights into the dual regulatory

capacity of this brain region, highlighting its importance in enhancing neural flexibility to optimize brain functions.

## Discussion

Our results demonstrate the significant modulatory effect of claustrum axons on dPFC neurons during sensory processing. The diversity of responses observed across the sensory, opto-responsive, and opto-boosted subpopulations underscores the complex role the claustrum plays in shaping neuronal dynamics. Notably, the opto-boosted neurons, which exhibited no response to visual or optogenetic stimuli alone but responded to their combination, highlight the claustrum's ability to selectively enhance neural responsiveness under certain conditions. This modulation, characterized by increased response variability and temporal dynamics, supports the hypothesis that the claustrum enables greater flexibility in sensory integration within the dPFC.

The differential response patterns between excitatory and inhibitory neurons further suggest that the claustrum's influence is not uniform across the dPFC network. The finding that opto-responsive cells exhibited a stronger prevalence of inhibitory responses aligns with previous reports of the claustrum's role in regulating inhibition within cortical networks[6,43]. The increased synchronization observed during the claustrum silencing experiment suggests that the loss of claustrum-mediated input leads to compensatory network adjustments, potentially to maintain functional coherence. This enhanced synchronization might reflect an intrinsic mechanism by which the dPFC preserves information processing capacity in the absence of normal claustral input.

Our findings suggest that claustrum input modulates dPFC neural variability and network coordination, maintaining a balance between flexibility and stability in cortical processing. When claustrum input is removed, we observed a relative increase in population synchrony, particularly in trained animals. This may reflect a default shift toward increased local synchronization in the absence of modulatory control, similar to compensatory network dynamics observed following disruption of other modulatory systems. For example, removal of cholinergic or noradrenergic input has been shown to increase cortical synchrony, reduce variability, and constrain dynamic range[44–46]. While such compensatory synchronization may transiently stabilize network activity, excessive synchrony is often associated with reduced information processing capacity, as seen during disengaged or sleep states. Thus, we propose that claustrum input helps prevent runaway synchronization and maintains network flexibility, allowing adaptive sensory processing and cognitive flexibility under varying behavioral demands.

Interestingly, training did not significantly alter the basic response properties of the identified subpopulations, suggesting that the claustrum's modulatory effects are robust against experience-dependent plasticity in the context of sensory integration. However, the observed increase in network correlation post-training implies that while the claustrum supports flexibility and variability in neural responses, learning processes drive a more coordinated network state. This dual capability of the claustrum—to promote both variability and synchronization—suggests that it operates as a key regulator of both flexibility and stability in the neural circuitry of the dPFC, facilitating adaptive sensory processing.

The differential effects of claustrum activation and silencing in naive versus trained networks reveal significant insights into its functional role. While both activation and silencing of claustrum axons seem to produce similar outcomes in naive animals, activation uniquely enhances cross-correlation in trained networks, whereas inhibition does not alter the trained network's dynamics. This suggests that the claustrum might exert an inhibitory influence under baseline conditions, as evidenced by the more pronounced inhibited responses in the opto-boosted subpopulation of naive animals, compared to

excited responses—a distinction not observed during silencing. These findings align with previous research, indicating that the claustrum may serve to diversify neural responses or avert excessive synchronization, which could otherwise lead to inflexible and less adaptable neural processing. When the claustrum's modulatory effect is diminished, there appears to be a compensatory increase in synchronization, likely aimed at preserving coherent processing. Such adaptability in response dynamics highlights the claustrum's pivotal role in enhancing neural flexibility, ensuring that the brain can swiftly adjust to varying conditions and demands.

This approach comes with certain considerations. While stimulating claustrum axons locally within the PFC offers a targeted method to assess direct pathway-specific effects, widefield optogenetic activation may not uniformly recruit all claustrum axons. The efficacy of activation likely depends on the spatial distribution and circuit engagement of individual axons within the targeted region[16,47]. Despite this limitation, this strategy remains the most effective way to isolate and causally probe the direct influence of claustral input on prefrontal circuits, minimizing confounding polysynaptic effects that would arise from upstream or distal stimulation sites.

Notably, there are inherent differences in the response kinetics between eOPN3 and ChrimsonR, which are two distinct optogenetic tools. ChrimsonR produces rapid depolarization and spiking upon light activation, with millisecond-scale onset and offset kinetics[36]. In contrast, eOPN3 induces a longer-lasting inhibitory effect via Gi-coupled hyperpolarization, with slower onset and offset kinetics that can persist beyond the light pulse itself[42,48]. These kinetic differences could influence the temporal precision of excitation versus inhibition and potentially affect the dynamics of PFC responses, particularly for rapid sensory processing. Despite these differences, we chose to use the same photostimulation protocol for both to maintain consistency in the experimental setup. This approach ensures that any observed effects are due to the biological interactions rather than variations in stimulation techniques. However, it also means that the results should be interpreted with an understanding of how these kinetic differences might impact the magnitude, timing, or persistence of neural responses. By standardizing the protocol, we aimed to clearly isolate the contributions of the CLA-PFC interactions, while acknowledging that the slower offset of eOPN3 inhibition may dampen post-stimulation dynamics differently than the fast ChrimsonR activation. This highlights the importance of methodological consistency while also recognizing the need to consider the intrinsic properties of distinct optogenetic actuators.

Another consideration of the current study is the temporal resolution inherent to GCaMP6s-based calcium imaging. The relatively slow decay kinetics of GCaMP6s may obscure brief inhibitory periods or rapid rebound excitation that can occur within hundreds of milliseconds, particularly during the optogenetic stimulation window[49]. In our experimental design, PMT shutters were closed during the 250 ms optogenetic stimulation period to avoid photodetector damage, and responsiveness was assessed in the post-stimulation window. As a result, very fast transient inhibitory effects or rebound firing immediately following light offset may not be fully captured in our imaging data. Nevertheless, our primary aim was to assess population-level modulation of sensory responsiveness, variability, and network coordination, which occur over longer timescales well suited to calcium imaging. Previous electrophysiological studies of claustrum function have demonstrated both inhibitory and excitatory influences on cortical activity[6,22], consistent with the bidirectional modulation we observe at the population level. Future studies incorporating high-temporal-resolution electrophysiology alongside imaging will be valuable for dissecting the precise timing of fast inhibitory-excitatory sequences following claustrum activation.

In this study, we employed widefield optogenetic stimulation of claustral axon terminals in cortex to probe the modulatory influence of

claustral input on cortical processing. While this approach enabled broad activation of claustral projections across large cortical areas, we acknowledge that it does not fully recapitulate the native spatio-temporal patterns of claustrum activity. Claustral projections arborize across both superficial and deep cortical layers, but widefield 1-photon stimulation primarily activates superficial layers, with only partial penetration into deeper layers due to light scattering and absorption[50,51]. Direct stimulation of claustrum cell bodies could, in principle, provide a more physiologically accurate recruitment of downstream targets. However, such an approach is technically challenging due to the elongated, narrow, and curved anatomical structure of the claustrum along the anterior-posterior axis[2,28,52], making it difficult to stimulate a representative population of claustrum neurons using traditional optical methods. Cell body stimulation would likely engage only a small, spatially localized subset of neurons, potentially limiting the generalizability of the results. Despite these limitations, terminal stimulation allowed us to investigate the functional capacity of claustral input to modulate cortical responses at the network level, complementing future work that will be needed to dissect the full spatial and functional specificity of claustrum-cortex interactions[53,54].

In conclusion, our findings underscore the claustrum's role in dynamically modulating neural activity in the dPFC, enhancing the flexibility and synchronization of neuronal responses. The differential effects of activation and silencing of claustrum axons reveal its bidirectional regulatory capacity, emphasizing its importance in optimizing brain function. Future studies should explore the specific mechanisms through which the claustrum exerts this dual control and investigate its role in more complex cognitive tasks that require the integration of multiple sensory modalities.

## Materials & methods

### Animals
Male and female C57BL/6 J or Nkx2.1Cre;Ai9 background mice were used in these experiments. Twenty four GCaMP6s transgenic mice (CamKIIa-tTa x B6;DBA-Tg(tetO-GCaMP6s)2Niell/J) were used for imaging of the claustrum axon activation and inhibition experiments. Additionally, five Nkx2.1-Cre x tdTomato were used for inhibitory neuron imaging experiments. Mice were between 5–11 weeks of age when surgery was performed.Animal experimentation was carried out in accordance with the guidelines and regulations of the UK Home Office (Animals in Scientific Procedures Act of 1986) and the University of Oxford Animal Welfare and Ethical Review Board.

### Stereotaxic surgery
Mice were prepared for imaging experiments through a single surgical session that included headplate implantation, cranial window placement, and viral injection. Initially, the mice were anesthetized with 3% isoflurane, placed in a heated stereotaxic frame, and given intraperitoneal injections of 5 mg/kg meloxicam (Metacam) and 0.1 mg/kg buprenorphine (Vetergesic). They were maintained under 1.5% isoflurane and kept warm on a 37 °C heating pad throughout the surgery. The scalp was sanitized with chlorhexidine gluconate and isopropyl alcohol (ChloraPrep), followed by the application of a local anesthetic (Bupivacaine) under the scalp. A midline incision was made in the scalp, which was then retracted to reveal the skull, leveled manually between the bregma and lambda landmarks. The site for the 4 mm cranial window was marked stereotaxically (AP: -1 mm to 3 mm; ML: either -1 mm to 3 mm or -3 mm to 1 mm from Bregma, depending on the side of injection). A circular craniotomy was drilled at this marked location, and the skull piece was removed after saline application. The dura mater on the injected side was carefully excised with regular saline washing.

To specifically label claustrum axons in the PFC, AAV.hSyn.-Cre.WPRE.hGH (Retro-cre, 0.70e + 13 gc/mL, 90 nL, Addgene #105553) was injected into the retrosplenial cortex (AP: -3.0, ML:0.5, DV:-1.0) of

all mice to label the retrosplenial cortex projecting claustrum cells. Then, we injected AAV5-Syn-FLEX-rc[ChrimsonR-tdTomato] (Flex-ChrimsonR, 1.20e + 13 gc/mL, 250 nL, Addgene #62723) was injected into the claustrum (AP: 1.0, ML: 3.4, DV:-2.7) for activation experiments and pAAV-hSyn1-SIO-eOPN3-mScarlet-WPRE (FLEX-eOPN3, 1.20e + 13 gc/mL, 250 nL, Addgene #125713) for axon silencing experiments. Injections were performed with a glass needle and automated nanoinjector (Nanoject II™ Drummond) at a rate of 100 nL/min. The needle remained in place for 10 min post-injection to promote viral diffusion.

A cranial window, consisting of a 4 mm coverslip attached to a 5 mm coverslip, was then inserted into the craniotomy and sealed with cyanoacrylate (VetBond) and dental cement. An aluminium headplate with an imaging well centred on the window was then secured in place with dental cement (Super-Bond C&B, Sun-Medical). Post-surgery, mice were moved to a fresh cage, provided with meloxicam jelly for pain relief, and allowed a three-week recovery period to achieve optimal viral expression before beginning experiments.

To selectively express opsins in claustrum neurons, we injected AAV-retro-Cre into the RSP, which projects specifically to the claustrum but not to adjacent regions such as the insula. This strategy allowed retrograde delivery of Cre recombinase to a well-defined claustrum population. Subsequently, we injected Cre-dependent opsins (ChrimsonR or eOPN3) directly into the claustrum to restrict expression to claustrum neurons that received retrograde Cre. This approach minimized off-target labeling of nearby non-claustral regions and ensured selective manipulation of claustrum-originating axons within PFC. The same targeting strategy has been extensively characterized in our previous work[16] and in other labs[2,3], where retrograde Cre delivery from RSP combined with Cre-dependent opsin expression in the claustrum reliably labeled a large population of glutamatergic claustrum projection neurons. This population includes neurons projecting both to RSP and medial prefrontal cortex, reflecting the known collateralization of claustrum projections.

### Awake head-fixed two-photon calcium imaging and wide field photostimulation
All two-photon imaging was performed using a two-photon microscope (Ultima 2pPlus, Bruker Corporation) controlled by Prairie View software (Bruker Corporation), and a femtosecond-pulsed, dispersion-corrected laser (Chameleon, Coherent). Total power was modulated with a Pockels cell (Conoptics) and maintained at or below 50 mW on a sample for all experiments. Imaging was performed using a Nikon 16 × 0.8NA water immersion lens. The lens was insulated from external light using a custom 3D-printed cone connected to a flexible rubber sleeve. A wavelength of 920 nm and 50 mW power on the sample was used for visualizing GCaMP6s. An imaging rate of 30 Hz and a 512 × 512 pixel square field of view (FOV) were used for all recordings.

**Visual stimulation.** Visual stimuli consisted of 250 ms light pulses delivered using a high-brightness 590 nm LED (LYCP7P-JRJT-1-0, Farnell, UK), controlled via a National Instruments data acquisition card and PackIO software, with custom MATLAB scripts generating the timing signals. The LED was positioned ~10 cm lateral and slightly anterior to the contralateral eye, at an angle of ~30–45 degrees relative to the midline to ensure direct illumination of the visual field corresponding to the recorded hemisphere. The LED was driven at 1 A (manufacturer-rated luminous flux: 56 lumens). No additional focusing optics were used.

**Optogenetic stimulation.** Widefield (1 P) optogenetic stimulation was performed concurrently with two-photon imaging using the widefield excitation module of the Bruker 2pPlus system. Orange light photo-stimulation (595 nm, 10 pulses of 25 ms at 40 Hz; total duration 250 ms) was delivered during 2 P imaging at 1.5 mW power at the

sample. For optogenetic activation, we used ChrimsonR, a red-shifted opsin effective for activation (or eOPN3 for inhibition) with 594–595 nm illumination[36]. Light power at the fiber tip was maintained at ~5–10 mW/mm², sufficient to reliably activate claustrum neurons across the targeted region. This stimulation protocol was validated in the previous work using the same targeting strategy, where robust activation of claustrum projections and downstream cortical modulation were demonstrated[16]. To prevent photomultiplier tube damage during photostimulation, high-speed shutters were triggered to close all PMTs during optogenetic activation. Triggers for both light stimulation (Thorlabs M595L3 LED with LEDD1B driver) and PMT shutters were controlled via PackIO software.

To confirm that our two-photon imaging at 920 nm did not unintentionally activate Chrimson, we relied on the original spectral characterization of Chrimson[36], which shows negligible activation at 920 nm. This is further supported by Packer et al.[55], who found no cross-activation of the more sensitive opsin C1V1 during 920 nm two-photon imaging. These recordings were conducted using a conservative imaging protocol (2–3 s imaging windows and ≥40 s inter-trial intervals), further reducing any possibility of opsin desensitization or tonic activation. Our primary dataset showed no systematic baseline shifts or activity drifts that would suggest 920 nm–induced Chrimson activation.

Following recovery from surgery, mice were first acclimated to head fixation under the two-photon microscope. ChrimsonR and eOPN3 expression in claustrum axons projecting to PFC was then assessed visually through the cranial window under the microscope. Animals in which no labeled axons could be detected were excluded from further experiments. Only animals with clear ChrimsonR- or eOPN3-expressing axons visible in PFC were included for optogenetic activation or inactivation experiments.

During each experiment, mice were first head-fixed under the microscope. Imaging was performed in an enclosed hood to minimize visual stimuli, and white noise was used to obscure extraneous sounds. The surface of the cranial window was levelled relative to the imaging plane using a tip-tilt stage (Thorlabs). During each imaging session, mice were presented with 30 randomly interleaved stimulus presentations with and without optostimulation separated by randomly generated 8–15 sec intertrial intervals (30 visual-only trials, 30 opto-only trials and 30 visual + opto trials). These 90 stimuli were randomly drawn. The order of each trial type was randomly generated each day. After all stimuli were delivered, a new FOV was then selected and the sensory stimulation and photostimulation were repeated.

## Behavioural training

Pavlovian conditioning was employed to associate a simple visual stimulus with a water reward, thereby familiarizing the animals with the visual cue prior to imaging. This approach was chosen to ensure that the visual stimulus had behavioral relevance without requiring complex training or active task performance during imaging. Associative learning paradigms like Pavlovian conditioning have been shown to robustly engage sensory, prefrontal, and subcortical circuits, including the claustrum and prefrontal cortex[56], even under passive viewing conditions. By using a passive associative learning paradigm, we aimed to enhance stimulus salience while minimizing variability related to task performance, motor responses, or differences in learning strategies[41].

Mice were presented with a 2-sec LED stimulus, delivered unilaterally to ensure consistent sensory exposure across animals. In 80% of trials, the LED presentation was followed by a water reward, while 20% of trials omitted reward to maintain engagement and stimulus salience. Animals underwent daily training sessions consisting of 150 trials per session, with inter-trial intervals (ITIs) randomly drawn from a uniform distribution between 8 and 15 sec. Training was conducted over 7–10 consecutive days until animals reliably associated the visual

stimulus with reward delivery. No active behavioral responses were required for reward delivery. Imaging sessions were subsequently performed in a passive viewing context, using the same visual stimulus but without reward delivery, allowing assessment of neural responses in a controlled, behaviorally relevant but motor-free condition.

To evaluate whether trial-by-trial fluctuations in arousal contributed to the observed neural responses, we included pupil radius—a well-established proxy for arousal—as a regressor in a generalized linear model (GLM) for each cell's response. The model included predictors for stimulus condition (visual, opto, visual+opto), pupil radius (z-scored within session), and trial number to account for slow drift. Across the population ($n = 12{,}487$ cells from 16 mice), pupil radius did not significantly explain additional variance in ΔF/F responses beyond stimulus condition (mean additional variance explained = $0.27\% \pm 0.06\%$ SEM). Only a small minority of cells (3.4%) showed significant modulation by pupil radius at $p < 0.05$ (False discovery rate corrected), and these were not spatially clustered or functionally distinct. Moreover, the presence or magnitude of optogenetic modulation (quantified by ΔF/F difference between visual and visual+opto conditions) was not systematically related to pupil size (Spearman's $\rho = 0.03$, $p = 0.21$). These results suggest that arousal fluctuations were not a primary driver of the stimulus-evoked or optogenetically modulated responses reported here.

## Perfusion and tissue sectioning

The presence of claustrum labeling in each animal was confirmed using post-mortem coronal slices, spanning from 1.5 mm anterior to bregma to 0 mm, as shown in Supplementary Fig. 1.1. Mice were deeply anesthetized with 5% isoflurane followed by an overdose of pentobarbital administered intraperitoneally. Subsequently, transcardial perfusion was performed using 0.01 M PBS, followed by fixation with 4% PFA. The brains were extracted and allowed to fix overnight in 4% PFA. After fixation, the brains were transferred to 0.01 M PBS and prepared for sectioning with a Leica VT1000S vibratome. Coronal slices, 100 μm in thickness, were obtained and either immediately mounted in 0.01 M PBS or stored in a tissue freezing solution (45% 0.01 M PBS, 30% ethylene glycol, 25% glycerol) at −20 °C for up to three years. Before mounting, the tissue was washed three times for 5 min each in 0.01 M PBS and then coverslipped. Whole-slice and claustrum images were captured at 16X magnification using a 2 P microscope equipped with a Coherent Vision-S laser, Bruker 2PPlus microscope, and a Nikon 16 × 0.8 NA objective.

## Data Analysis

All analyses were performed with custom routines using Python 3.7.9 and open source packages unless otherwise stated.

**2-photon calcium imaging analysis.** Calcium imaging data were preprocessed using Suite2P[57] to remove motion artifacts and cell segmentation. We computed ΔF/F for each cell using the equation:

$$\Delta F/F = (F - \underline{F})/\underline{F}$$

where $\underline{F}$ represents the mean of $F$ across time through the entire session. For cell ROIs selected by suite2P, $F$ was first corrected for neuropil fluorescence by subtracting 0.7*FNeu. After calcium traces were exported from Suite2P, all analyses were carried out using custom Python and Matlab code.

Extracted calcium signals were then analyzed to identify cells that significantly responded to visual, opto or visual + opto stimulation. Significantly responsive cells were identified by using a non-parametric Mann-Whitney U test to compare the signal in the 1 sec before and after stimulus onset. Multiple comparisons correction was performed using the Benjamini-Hochberg false discovery rate analysis with an alpha of 1%.

**Event time extractions.** Synchronised timeseries data signals collected as individual channels in PackIO (visual cues and photostimulation) were processed with custom code. Widefield photostimulation timings were retrieved from the high-speed shutter loopback signal collected as a temporally synchronised PackIO channel. The photostimulation trigger timestamp and the total duration of the photostimulation protocol were used to define each photostimulation trial's onset and offset. The photostimulation onset/offset timestamps were cross-referenced to the imaging frame clock signal to define photostimulation onset and offset in the imaging data. We excluded imaging frames between onset and offset of each photostimulation trial as photostimulation laser induced imaging artefacts and aberrant neuronal activity.

To quantify the intricacies of neuronal activity, we used four different metrics, absolute magnitude, coefficient of variation (CV), dynamic time warping (DTW), and empirical cumulative distribution function (ECDF). The absolute magnitude is calculated to determine the intensity of response changes by comparing mean fluorescence levels before and after a stimulus. This measure, by accounting for noise through standard deviation, aims to quantify signal changes from background fluctuations. To quantify trial-to-trial variability of neural responses, we computed the coefficient of variation (CV) based on trial-averaged response amplitudes. For each trial, we extracted fluorescence signals ($\Delta F/F$) for a 1-sec window aligned to stimulus offset and calculated the mean fluorescence across all frames within this window, resulting in a single response amplitude per trial. We then computed the CV for each cell as the standard deviation of these trial-averaged responses divided by their mean across trials. This approach allowed us to capture variability in the overall response magnitude across repeated presentations while minimizing the influence of frame-to-frame fluctuations inherent to calcium imaging signals.

To assess trial-to-trial variability in the temporal dynamics of neural responses, we employed dynamic time warping (DTW), a well-established method for aligning time series with temporal shifts or distortions[58–60]. For each cell, fluorescence signals ($\Delta F/F$) were extracted for a 1-sec window aligned to stimulus onset across all trials. DTW was then used to compute pairwise distances between the temporal response trajectories of all trial pairs for each cell. The DTW algorithm allows flexible nonlinear alignment of time series, accounting for small variations in response onset, peak latency, or shape across trials. For each cell, we quantified temporal variability as the average DTW distance across all trial pairs. This approach captures differences in temporal structure of calcium responses that may not be reflected by amplitude-based measures alone, providing a complementary index of response consistency across trials.

Finally, The ECDF further by illustrating the cumulative distribution of fluorescence data, allows us to visualize and interpret the variability and distribution of responses across different conditions. In this study, we leveraged our extensive dataset (>40,000 cells across animals) to examine population-level shifts in $\Delta F/F$ responses between experimental conditions. For each animal, we computed the mean $\Delta F/F$ response of individual cells within a fixed window following stimulus onset, and then compiled these values into condition-specific distributions.

To emphasize the structure of the bulk population and minimize the influence of outliers—which are common in large-scale imaging datasets—we focused our ECDF analysis on the central 90% of the distribution, defined between the 5th and 95th percentiles. This approach offers a more stable and representative view of population-level dynamics, preserving meaningful variability while reducing distortion from rare extreme values. By using this robust, non-parametric method, we were able to detect condition-dependent distributional shifts that reflect how widespread neuronal populations are modulated, beyond mean or median response summaries.

Together, these calculations enable a comprehensive analysis of the data, highlighting both the magnitude and variability of neuronal responses as well as their temporal coordination.

## Reporting summary

Further information on research design is available in the Nature Portfolio Reporting Summary linked to this article.

## Data availability

All processed data deposited in the Figshare (https://doi.org/10.6084/m9.figshare.30164857.v2). Any additional data required to reanalyze the data reported in this work paper is available from the lead contact upon request.

## Code availability

All codes and functions used to analyse and generate the figure panels in this study are available in GitHub (https://github.com/huriyeatg/clapfcstimulation). Any additional information required to reanalyze the data reported in this work paper is available from the lead contact upon request.

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

## Acknowledgements

This work is supported by a Sir Henry Wellcome postdoc fellowship (222807/Z/21/Z) to H. A., grants from Wellcome (213465/Z/18/Z) and the European Research Council (ERC) under the European Union's Horizon 2020 research and innovation programme (grant agreement No 852765) to A.M.P. The authors gratefully acknowledge Rob Lees, Andrew M.

Shelton and Armin Lak for their insights and discussion in the completion of this project.

## Author contributions

H.A. conceived the study, designed and performed all experiments, analyzed the data, and wrote the manuscript. I.P.L. assisted with data collection for the silencing experiments. A.M.P. contributed to experimental design, provided input on data analysis, assisted in manuscript preparation, and secured funding & laboratory resources.

## Competing interests

The authors declare no competing interests.
