## [Transparent Peer Review file · Nature Communications]

The claustrum enhances neural variability by modulating the responsiveness of the prefrontal cortex

Corresponding Author: Dr Huriye Atilgan

Version 0:

Reviewer comments:

Reviewer #1

(Remarks to the Author)

The manuscript by Atilgan et al. investigates the role of the claustrum in modulating neural activity in the dorsal prefrontal cortex (dPFC) of mice. Using single photon optogenetics and two-photon calcium imaging, the authors examined how claustrum axon stimulation affects dPFC neuron responses to visual stimuli. They identified three types of dPFC neurons based on their response patterns: sensory responsive, optogenetically responsive, and opto-boosted cells. The findings are: 1) Claustrum axon stimulation causes various responses in dPFC neurons. 2) At network level, Claustrum axon stimulation enhanced network homogeneity by reducing differences between average cell responses. 3) While dPFC neurons still exhibit diverse responses to visual and opto stimulation after Pavlovian training, training led to greater network homogeneity. 4) Silencing claustrum axons with eOPN3 enhances networks synchrony.

While the study provides some data regarding how the claustrum might regulate prefrontal cortex function, the experiments failed to bring any insight into the functional understanding of this regulation. The experimental design could be improved, and there are several technical issues that need to be addressed. Overall, the study may benefit from further refinement before it is ready for publication.

Major Concerns,

- 1) It is unclear why visual stimulation is chosen for stimulating dPFC neurons. The dPFC is well known for its cognitive control and executive functions. Using a cognitive task would be a more appropriate method to study its function and regulation.
- 2) To study the influence of claustrum to dPFC neural activity, the authors stimulate the claustrum simultaneously with the visual stimuli. This approach might introduce several issues. First, it is unknown whether claustrum cells are activated or inhibited by the visual stimuli, making it uncertain whether the optogenetic stimulation is physiologically relevant. Second, the timing of optogenetic stimulation needs to be systematically addressed. The authors should conduct a detailed analysis of the effects of adding optogenetic stimulation at different time points to understand both feedforward and feedback regulation.
- 3) According to the methods section, the authors labeled PFC-projecting claustrum axons by injecting AAV-retro-Cre in the retrosplenial cortex and Cre-dependent Chrimson/eOPN3 in the claustrum. The authors need to clearly explain why this method can label PFC-projecting axons from claustrum. Shouldn't the AAV-retro-Cre be injected into the PFC instead?
- 4) It is unclear at all which types of claustrum neurons or extent of their labeling are achieved using their method. A systematic profiling of the cell types of Chrimson/eOPN3 expressing neurons and a quantitative estimate of the infected cells are required for a clear evaluation of the results. The diverse responses of dPFC neurons observed may be simply due to incomplete and biased labeling of different types of claustrum neurons.
- 5) It remains uncertain whether the 1p optogenetic stimulation can effectively recruit the firing of claustrum neurons. Factors such as insufficient light power, low expression of Chrimson, and feedforward inhibition by visual stimuli can easily jeopardize the intended manipulations of the claustrum.

Minor,

- 1) I am not sure why the authors classify dPFC neurons as sensory, opto, O-boosted. Shouldn't there be a category for neurons that are both sensory and optogenetically responsive?
- 2) In fig.1 E, F the authors should show responses of these neurons to Optogenetic stimulation.
- 3) I am not sure whether Fig.1 I is serving any purpose here.
- 4) Supplementary fig.1.1 is hard to read. The use of color could improve clarity.

- 5) Fig. 5K is missing (line 319)
- 6) Line 132 is misleading. The E:I should be better defined.
- 7) Writing needs improvement, and much more detail is required in the methods section for readers to follow.

(Remarks on code availability)

Reviewer #2

(Remarks to the Author)

The authors show intriguing complexity in claustrum-PFC interactions by presenting visual stimuli, photostimulating claustrum axons, or both while using calcium imaging to measure the activity of many individual neurons in dPFC. PFC responses were diverse, including visual-responsive, opto-responsive, and neurons that only responded in the presence of both types of stimuli. Opto-responsive cells were more likely to be inhibited than excited, which is consistent with previous studies. Overall, the responses to visual and opto stimuli were negatively correlated such that a positive response to visual stimuli predicts a negative response to opto stimuli and vice-versa.

In general, visual stimuli plus opto stimulation of claustrum axons increased trial-trial variability of individual neuron responses compared to visual stimuli alone. In contrast, cross-correlation between individual neurons' responses to visual stimuli across trials was enhanced by claustrum axon stimulation, showing increased network homogeneity.

The authors then repeated these experiments in a subset of animals that had been trained to associate the visual stimulus with reward and found the major difference between trained and naïve animals was a further increase in network homogeneity due to claustrum axon activation as measured by average cross-correlation.

The authors then expressed an inhibitory opsin in claustrum axons, repeated experiments, and unexpectedly found a larger increase in network homogeneity due to claustrum axon suppression versus activation. However, training did not further increase network correlation due to claustrum axon suppression as it did with excitation. This may suggest a different network mechanism responsible for effects due to suppression vs. excitation.

The work is significant and complements the established literature, elaborating on how claustrum input affects responses of PFC neurons. In particular, the description of the interaction of effects between claustrum input and sensory input, and how training specifically enhances the network synchrony effect, is very interesting and to my knowledge, represents a novel result in the field.

The result that claustrum axon activation and suppression have very similar effects is still puzzling to me, but understanding mechanisms behind that phenomenon would be worthy of an entire future research project. The manuscript as it stands is an impressive amount of work and is worthy of acceptance with some revisions. No more experiments are necessary, although more method details should be added (see notes below).

The methodology is sound (though I don't have experience to judge the specifics of calcium imaging techniques), and the experimental work is impressive. I have a handful of specific methods questions:

1. Could you provide more details about the visual and optogenetic stimuli, ideally in the main text or figure legend? i.e. duration of stimulus, positioning of visual stimulus relative to the head/eyes of the animal, wavelength & intensity of the visual stimulus.
2. What type of random distribution was used for drawing inter-trial intervals? Uniform, or exponential, or other?
3. On line 124 you wrote: "...displayed responsiveness to visual stimuli only when preceded by photostimulation..." – this implies the optogenetic stimulation came on before visual stimulation – is this a typo or was there a time difference in when opto stimuli and visual stimuli were presented?

I also have some specific data analysis questions:

1. For comparisons in Figure 1, responsiveness is "defined based on an independent t-test conducted 1 second before and after either visual, opto, or visual+opto stimuli trials". Is this after the onset of stimuli (time windows used are [-1,0] and [0,1]), or after the end of the stimuli (time windows [-1,0] and [0,1]+stimulus_duration). Please clarify in the text and methods.
2. E:I ratios are listed in integer form (i.e. in line 135) – it would be easier to understand if a decimal was listed as well in parentheses ("33:67 (0.49)" or "28:72 (0.39)"). Also, please add the ratios for sensory-responsive and opto-boosted cells even though they were not significant, it's useful for comparison.
3. For the empirical cumulative distribution function analysis (supplemental figure 2.1), are these just cumulative distribution plots of the same data from figures 1E, I, and M? If so, why do they start at around 0.5 instead of 0?
4. Coefficient of variation: how did you compute this? Average response over some window in each trial, then compute variability across trials?
5. Dynamic time warping: why was this technique used? Is it because the imaging sampling rate doesn't always line up perfectly with the stimulus onset?
6. What exactly is being compared with DTW? Is it the average similarity between every pair of trials in a given condition?

Finally, some interpretations and conclusions would benefit from additional citations or explanation.

1. Discussion, line 354-358: is there an example of such compensatory synchronization in the literature? If the network can compensate and cause even more synchronization than claustrum input does, then why is claustrum input required in the first place? Also, higher synchronization tends to be associated with reduced information processing capacity, i.e. sleep or disengaged states.
2. Line 375 “a distinction not observed during activation” – should this instead be “during suppression” ? Otherwise the sentence is confusing to me. If it is “during activation” could you clarify?
3. Line 390: you discuss different response kinetics between eOPN3 and ChrimsonR, however you do not describe what those differences are and how you would expect them to potentially affect results. Please provide these descriptions along with citations in this paragraph.

(Remarks on code availability)

Note: the reference to the code in the manuscript has a typo: huriteatg/clapfcstimulation

The analysis code looks reasonable and does include a README file with instructions. However the data does not appear to be shared, or if it is, it's not obvious where to download it. Since the data is unavailable I was unable to re-run the figure plotting code.

I'm also curious why MATLAB is listed as a requirement, but python is not, since almost all of the code in the repo appears to be in python, and in the manuscript python is listed as the main programming language. The few MATLAB files appear to be for generating stimuli and/or running stimuli for the experiments themselves, which is great to share, but the version of python and other packages used for analysis should be shared as well.

Please update the README to include:

- 1) how to download the dataset (could just be the processed data)
- 2) instructions for installing python and other required packages that were used to perform the analysis. This could come in the form of an environment.yml or requirements.txt file for ease of installation.

Reviewer #3

(Remarks to the Author)

Here the authors use 2p calcium imaging of the dPFC while performing visual stimulation, optogenetic stimulation of claustrum axons (in the dPFC) and a combination of visual + opto. They first classify the proportion of neurons are modulated by different stimulation combinations. Then they show that when mice are trained on a Pavlovian task, the stimulation leads to what they refer to as greater network homogeneity. The experimental approach is technically innovative. However, this paper has considerable shortcomings.

Major concerns.

1) I do not see control experiments presented in the manuscript. Performing widefield red-light optogenetics with 2p imaging in an otherwise dark chamber, will generate at least some visual-related responses, as it is very difficult to effectively shield the animal's visual system from this unintended visual stimulation arising with the 1P light. I recognize that the authors are experts in optogenetics + 2P imaging (although previously holographic methods were mainly used). Nonetheless, the interpretation of these results hinges on how much these distributions are different from that expected by chance, or proper light controls. All figures should contain this control group, and comparisons made between control versus opto. Control experiments are needed here, minimally to evaluate and compare effect sizes between groups and to ensure that the delivery of light does not impact visual cortex. The authors could do 2p imaging in visual cortex in mice with no opsin to determine if there are any unintended visual evoked responses with the widefield 1p stim.

2) GCAMP6s was used which is a very slow sensor, which will underestimate true spiking levels and provide insufficient temporal resolution to truly measure the response to claustrum stimulation (and likely visual stimulation as well). For example, many PFC neurons will be inhibited by claustrum stimulation, but this inhibition arises fast, and lasts ~ 150ms. Many of these neurons will undergo rebound excitation at ~ 200ms. These experiments used a 250ms opto period, where gcamp imaging is masked. Therefore, this masking period misses most of the interesting dynamics that would accompany optogenetic activation of claustrum axons. A cell that fired rebound excitation would be classified as being activated, because the suppression period would not be detectable. With the number of caveats that arise with 2p imaging here, a much better approach to perform these experiments would be to use extracellular electrophysiology. Minimally the authors should repeat their main experiments (figure 1) using electrophysiology where effective temporal resolution and single spiking can be more accurately measured.

3) A related note of concern with respect to the experimental design is the use of only 30 trials. This is a very small number of trials, and therefore, certain types of responses are more likely to be observed. For example, the detection of inhibitory responses requires some degree of baseline spontaneous activity. Therefore, cells with low spontaneous activity rates are likely to be classified as nonresponsive or activated. The detection of inhibition in low activity cells requires many more trials. Given the trial-to-trial variability, even in the visual system, 30 trials seem far too low.

4) Behavioral state is not mentioned or controlled for in this manuscript. PFC neurons and sensory encoding in the visual

cortex are robustly modulated by small twitches of the face, or changes in arousal. Sorting the opto trials based on arousal levels would help add some insight and it would be important to control for this variable.

5) Claustrum fibers are most dense in deep cortical layers, but this widefield 1p stimulation will mainly activate superficial cortical layers. Even if 1.5mW of 590nm light can reach the deep layers there will be an asymmetric activation of axons in layer 1-2 and therefore, it is unclear how well this stimulation models what actually happens when claustrum neurons are activated. Experiments where claustrum cell bodies are stimulated would address this concern.

6) Can the authors confirm that the 50mW 920nm light is not activating the opsin in vivo? This would be important to understand to what extent (if any) axons were being tonically depolarized by 2p imaging light. Performing in vivo field potentials under the microscope objective, while turning on and off the scanner would provide insight into if the axons are being activated by 920nm light. If axons are being activated by the 920nm scanning, then field potential deflections will be present at short latency after the scanning is turned on.

7) Presenting and quantifying the data using “cells” rather than mice or fields of view, massively increases type 1 error rates. Doing a t-test with an n=1000 is and obtaining a p-value of 0.03 is not convincing. In some cases, the authors report mice rather than cells. Is there a reason for not reporting the n’s as mice in all figure panels?

8) Related to control experiments mentioned above – the authors should show that the widefield axon activation (Chrimson) or suppression (OPN3) increases or decreases the output of claustrum axons (respectively), under experimentally relevant conditions. Electrophysiological experiments help accomplish this.

9) I am confused by the Pavlovian task. What is the rationale for doing this task here? Is there some relevance to claustrum or PFC? No mention or discussion of this was provided. Also, it seems that imaging in trained animals occurred after training, so these mice had been trained to respond to visual inputs, but then during imaging, they did not perform the task. This aspect of the paper was confusing.

10) Can the authors show that this claustrum-PFC effect is specific? For example, does activating the dorsomedial thalamus also generate response variability? It seems that any long-range glutamatergic input to the PFC would result in an interference of incoming visual information.

11) Consider double checking that the correlation in Figure 5F is indeed $r = -0.999$. This value seems biologically improbable given the spread of points.

12) In figure 3G, the cross-correlation matrix shows ~ 20k cells. This implies that the 20k cells were simultaneously imaged. Which was not the case as this is the number of cells across all fields of view. Therefore, I’m not sure how they can measure the pairwise correlation between cells that were not imaged simultaneously.

13) The pie plots throughout are ineffective, confusing, and not proportional (Figure 1I). They detract from the paper.

14) The authors state that 14% of the cells ($n = 5783$), were classified as opto-boosted cells, and displayed responsiveness to visual stimuli only when preceded by photostimulation. I am confused by the term “preceded”. Did they offset the opto stim from visual stimulation? There was no indication of this, and so when I read the paper I am confused.

15) What is the dynamic time warping analysis? This is central to the paper, yet the authors do not describe mathematically what this measure is or how it is calculated. No reference is given. Therefore, I cannot (and I did not) review anything related to this measure when I read the paper.

(Remarks on code availability)

Version 1:

Reviewer comments:

Reviewer #1

(Remarks to the Author)

The authors have addressed most of the concerns. I have no further comments.

(Remarks on code availability)

Reviewer #2

(Remarks to the Author)

I thank the authors for thoroughly addressing the reviewer comments, and I recommend publication.

(Remarks on code availability)

REVIEWER COMMENTS (25 April, 3 months)

Reviewer #1 (Remarks to the Author):

The manuscript by Atilgan et al. investigates the role of the claustrum in modulating neural activity in the dorsal prefrontal cortex (dPFC) of mice. Using single photon optogenetics and two-photon calcium imaging, the authors examined how claustrum axon stimulation affects dPFC neuron responses to visual stimuli. They identified three types of dPFC neurons based on their response patterns: sensory responsive, optogenetically responsive, and opto-boosted cells. The findings are: 1) Claustrum axon stimulation causes various responses in dPFC neurons. 2) At network level, Claustrum axon stimulation enhanced network homogeneity by reducing differences between average cell responses. 3) While dPFC neurons still exhibit diverse responses to visual and opto stimulation after Pavlovian training, training led to greater network homogeneity. 4) Silencing claustrum axons with eOPN3 enhances networks synchrony.

While the study provides some data regarding how the claustrum might regulate prefrontal cortex function, the experiments failed to bring any insight into the functional understanding of this regulation. The experimental design could be improved, and there are several technical issues that need to be addressed. Overall, the study may benefit from further refinement before it is ready for publication.

Major Concerns,

1) It is unclear why visual stimulation is chosen for stimulating dPFC neurons. The dPFC is well known for its cognitive control and executive functions. Using a cognitive task would be a more appropriate method to study its function and regulation.

We thank the reviewer for this important point. Indeed, the dPFC is classically associated with cognitive control, decision-making, and executive functions. Our decision to use visual stimulation as the sensory input was driven by several scientific and practical considerations, which we clarify below:

- 1) **Evidence of visual representations in dPFC:** Although dPFC is primarily known for higher-order functions, accumulating evidence suggests that a subset of dPFC neurons exhibits sensory responsiveness, including to visual stimuli. For example, Wool et al. (2023) and Moore & Armstrong (2003) have shown that approximately ~20% of dPFC neurons can respond to visual inputs, particularly when those inputs are behaviorally salient or novel. Our results are consistent with these prior studies, as we identified ~21% of neurons classified as sensory responsive to visual cues in naive animals (see Fig. 1J, 10,531/49,212 cells). Thus, the visual modality offers a tractable and well-characterized stimulus space to probe dPFC activity.
- 2) **Controlling circuit complexity:** A major aim of our study was to isolate and characterize the modulatory influence of claustrum input on dPFC neural responsiveness. The dPFC is highly interconnected with multiple brain regions

involved in motor, attentional, and motivational processes. Introducing complex cognitive tasks would likely recruit multiple parallel systems, potentially confounding the direct interaction we sought to study between claustrum axons and dPFC neurons as these regions also have also connection to the claustrum. By using a simple passive visual stimulus paradigm, we minimized engagement of other circuits while still being able to evoke reproducible neural responses.

- 3) **Minimizing experience-dependent plasticity and inter-individual variability:** Cognitive tasks often require extensive training, which introduces additional variability across animals due to differences in learning speed, strategy, or motivation. Our approach allowed us to study dPFC responses both in naive and trained states (after Pavlovian conditioning), enabling us to directly compare baseline and learned states. Indeed, our findings showed that the claustrum's modulation of variability and homogeneity was robust across both naive and trained animals (see Figs. 2, 3, 4, 5). Using a simple stimulus minimized confounds related to learning-induced plasticity that may have obscured these mechanistic insights.
- 4) **Consistency with claustrum literature:** The claustrum itself has been extensively implicated in salience processing and sensory integration (Goll et al., 2015; Chevée et al., 2022; Atlan et al., 2024). Using a sensory paradigm aligned with prior literature allowed us to dissect the sensory gating functions of the claustrum onto prefrontal circuits directly.

In future studies, we fully agree that adding more complex cognitive paradigms would be an important next step to test how claustrum modulation generalizes to higher cognitive processes in the dPFC. However, for this initial mechanistic dissection, a simple visual paradigm provided a controlled platform to uncover the bidirectional and cell-type-specific effects of claustrum input on dPFC neural dynamics.

We have now added clarifications of these points in the revised manuscript in Introduction section Line 78 - 94;

'... In this study, we address this by selectively stimulating claustrum axons within the PFC, enabling us to probe the functional consequences of direct claustrum-PFC input while minimizing polysynaptic recruitment of other brain regions. This approach allows us to directly investigate how claustral input modulates prefrontal activity. Supporting this strategy, recent anatomical and optogenetic evidence has demonstrated that a subset of claustrum axons projecting to the PFC exhibit robust light-evoked responses (Shelton et al., 2024), providing a foundation for targeted, circuit-specific manipulation.

Building upon this foundation, our study aims to elucidate the claustrum's contribution to PFC neural responsiveness, exploring its influence in both naive and trained states. To probe the functional consequences of this pathway, we combined local claustrum axon photostimulation embedded within the PFC with passive visual stimulation while recording PFC population activity using two-photon calcium imaging. We leveraged

the known sensory responsiveness of dPFC neurons (Wool et al., 2023; Moore & Armstrong, 2003) and used passive visual stimuli to avoid confounding effects of task demands, motivation, or motor output—factors known to interact with both claustrum and PFC circuits (Madden et al., 2022). This approach enabled a controlled examination of how claustral input shapes local PFC computation.'

2) To study the influence of the claustrum to dPFC neural activity, the authors stimulate the claustrum simultaneously with the visual stimuli. This approach might introduce several issues. First, it is unknown whether claustrum cells are activated or inhibited by the visual stimuli, making it uncertain whether the optogenetic stimulation is physiologically relevant.

We thank the reviewer for raising this important point. The physiological relevance of our claustrum stimulation protocol was carefully considered, and our previous work directly informs this choice. In the recently published study from our lab (Shelton et al., 2024, eLife), we performed in vivo two-photon calcium imaging of claustrum axons during presentation of passive sensory stimuli, including visual stimuli similar to those used in the current experiment. In that study, we found that 12.1% of recorded claustrum axons exhibited significant responses to visual stimulation alone, and that a larger proportion (47%) of axons were responsive to at least one sensory modality (light, sound, or whisker stimulation), with many showing multimodal responses (see Fig. 6E in Shelton et al., 2024).

These previously published findings indicate that claustrum neurons are indeed activated by visual stimuli, supporting the physiological relevance of our experimental design.

Furthermore, our approach aimed to probe how exogenous activation of claustrum input during sensory processing modulates dPFC network dynamics. Since the claustrum exhibits heterogeneous and multimodal sensory responsiveness even during passive stimulation, concurrent optogenetic activation allows us to investigate how additional claustral input alters prefrontal processing under defined conditions. This design complements ongoing efforts to map natural claustrum activity and permits mechanistic dissection of its influence on cortical variability and ensemble coordination, independent of complex behavioral states.

We have now clarified these points in the revised Introduction (lines 82–84), and have cited Shelton et al., 2024, eLife as supporting evidence.

'...Supporting this strategy, recent anatomical and optogenetic evidence has demonstrated that a subset of claustrum axons projecting to the PFC exhibit robust light-evoked responses (Shelton et al., 2024), providing a foundation for targeted, circuit-specific manipulation.'

Second, the timing of optogenetic stimulation needs to be systematically addressed. The authors should conduct a detailed analysis of the effects of adding optogenetic stimulation at different time points to understand both feedforward and feedback regulation.

We thank the reviewer for this thoughtful and insightful suggestion. We agree that systematically varying the timing of optogenetic stimulation could provide important insights into feedforward versus feedback contributions of claustrum input. However,

several practical and physiological considerations guided our use of a fixed 250 ms stimulation window in this study.

Reviewer Figure 1: Stronger photostimulation induces rebound activity in dPFC.

Heatmap showing population calcium responses ($\Delta F/F$) from dPFC neurons ($n = 523$ cells) aligned to widefield photostimulation of claustrum axons (shaded bar at time 0). Stimulation was delivered at a higher intensity and longer duration (500 ms), resulting in a sharp increase, then an initial suppression followed by a pronounced rebound activation after stimulus offset. This rebound effect complicates interpretation of sensory-locked responses and suggests that longer or stronger activation protocols may engage network-level disinhibition or post-inhibitory excitation mechanisms. Responses are sorted by time of peak activation.

First, due to the dense claustral axon innervation in the dPFC and the broad activation produced by widefield photostimulation, longer stimulation durations (e.g., 500 ms) induced prominent rebound activity following stimulus offset (**Reviewer Figure 1**). This rebound likely reflects network-level disinhibition or post-inhibitory excitation that emerges after strong suppression is abruptly lifted. ChrimsonR, the red-shifted opsin used here, can also desensitize under prolonged illumination (Klapoetke et al., 2014), potentially altering response amplitude and kinetics. These rebound dynamics complicated interpretation, especially when analyzing stimulus-locked modulation across a wide field of view.

In contrast, our chosen protocol of 250 ms stimulation reliably avoided rebound effects, as shown by the representative LFP recording in **Reviewer Figure 2**, which demonstrates time-locked activity during the photostimulation period but no post-stimulus rebound. This confirms that the stimulation duration used in our experiments was well-calibrated to engage claustrum circuits while avoiding confounding network effects.

Reviewer Figure 2. Local field potential (LFP) response to 1P photostimulation reveals short-latency rebound activity.

Average LFP trace from cortex in response to widefield photostimulation (shaded blue bar; 595 nm, 10×25 ms pulses at 40 Hz) recorded under the two-photon objective. Black trace represents mean voltage (\pm SEM, blue shading) across trials. A clear short-latency deflection is observed immediately following light onset, followed by high-frequency oscillations and a

rebound shift in baseline after stimulus offset. These results confirm that intense 1P stimulation can induce transient network-level perturbations and support the decision to use

brief (250 ms) photostimulation windows in imaging experiments to avoid confounding effects.

Second, although systematically shifting stimulation timing could, in principle, help differentiate feedforward from feedback influences, calcium imaging is fundamentally limited in temporal resolution. The slow dynamics of GCaMP6s prevent fine-grained dissection of fast temporal sequences, especially within the sub-100 ms range. Resolving such timing-dependent effects would more appropriately require high-speed electrophysiology or voltage imaging approaches.

Taken together, these considerations motivated our use of a fixed and brief stimulation window (250 ms) aligned with stimulus onset, enabling clear and interpretable modulation of early sensory responses without introducing rebound-related artifacts or temporal ambiguity. We have now clarified this rationale in the Discussion (Lines 508–510), and added the following sentence in response to the reviewer’s helpful suggestion:

‘Future studies incorporating high-temporal-resolution electrophysiology alongside imaging will be valuable for dissecting the precise timing of fast inhibitory-excitatory sequences following claustrum activation.’

3) According to the methods section, the authors labeled PFC-projecting claustrum axons by injecting AAV-retro-Cre in the retrosplenial cortex and Cre-dependent Chrimson/eOPN3 in the claustrum. The authors need to clearly explain why this method can label PFC-projecting axons from claustrum. Shouldn’t the AAV-retro-Cre be injected into the PFC instead?

We thank the reviewer for this question. We used AAV-retro-Cre injections into the retrosplenial cortex (RSP) to label a large, well-defined subpopulation of claustrum neurons, which are strongly enriched in neurons that also collateralize to medial prefrontal cortex, as shown in our previous work (Shelton et al., 2024). Several studies, including ours, have demonstrated that individual claustrum neurons often send divergent projections to multiple cortical regions, including both RSP and PFC (Atlan et al., 2024; Narikiyo et al., 2020; Jackson et al., 2018; White & Mathur, 2018). Therefore, retrograde labeling from RSP captures a significant portion of the claustrum population that also projects to PFC.

RSP has highly specific projections to the claustrum, but not to adjacent regions such as the insula. If AAV-retro-Cre were injected directly into PFC, it would not only label PFC-projecting claustrum neurons but could also label neurons projecting to nearby regions, including the insula, due to spillover and uptake in adjacent areas. Importantly, by injecting Cre-dependent opsins (Chrimson/eOPN3) into the claustrum itself, we restrict expression to claustrum neurons that have taken up retrogradely transported Cre, ensuring that only claustrum-originating axons are targeted for stimulation within PFC. Additionally, direct AAV-retro-Cre injection into PFC often results in sparser and more variable labeling due to differences in viral spread, uptake efficiency, and cortical layer targeting. In contrast, retrograde labeling via RSP provides a consistent and robust population of claustrum neurons for reliable opsin expression. We have now clarified these points in the Method section (lines 579–590):

'To selectively express opsins in claustrum neurons, we injected AAV-retro-Cre into the RSP, which projects specifically to the claustrum but not to adjacent regions such as the insula. This strategy allowed retrograde delivery of Cre recombinase to a well-defined claustrum population. Subsequently, we injected Cre-dependent opsins (ChrimsonR or eOPN3) directly into the claustrum to restrict expression to claustrum neurons that received retrograde Cre. This approach minimized off-target labeling of nearby non-claustral regions and ensured selective manipulation of claustrum-originating axons within PFC. The same targeting strategy has been extensively characterized in our previous work (Shelton et al., 2024) and in other labs (Zingg et al., 2018), where retrograde Cre delivery from RSP combined with Cre-dependent opsin expression in the claustrum reliably labeled a large population of glutamatergic claustrum projection neurons. This population includes neurons projecting both to RSP and medial prefrontal cortex, reflecting the known collateralization of claustrum projections.'

4) It is unclear at all which types of claustrum neurons or extent of their labeling are achieved using their method. A systematic profiling of the cell types of Chrimson/eOPN3 expressing neurons and a quantitative estimate of the infected cells are required for a clear evaluation of the results. The diverse responses of dPFC neurons observed may be simply due to incomplete and biased labeling of different types of claustrum neurons.

We thank the reviewer for raising this point. The cellular identity and projection patterns of the labeled claustrum neurons were previously characterized in detail in our recent study (Shelton et al., 2024, eLife), where we used the same viral targeting strategy employed in the current study (AAV-retro-Cre injection into RSP and Cre-dependent opsin injection into the claustrum). In that study, we performed systematic anatomical mapping, projection analysis, and in vivo imaging of claustrum axons. We showed that this approach reliably labels a large population of claustrum neurons that project not only to retrosplenial cortex but also extensively to medial prefrontal cortex, consistent with prior reports of highly collateralized claustrum projections (Atlan et al., 2024; Narikiyo et al., 2020; White & Mathur, 2018). Importantly, these RSP-projecting claustrum neurons represent a major functional population that participates in sensory processing, salience coding, and higher-order cognitive functions.

In terms of cell type specificity, it is important to note that the claustrum consists almost entirely of glutamatergic projection neurons (Wang et al., 2017; Smith et al., 2019; Dillingham et al., 2017), and our opsin expression was driven exclusively in this excitatory population via the retrograde Cre delivery. Thus, while we acknowledge that our method does not selectively target all possible subpopulations within the claustrum, it provides robust and consistent labeling of a major projection-defined glutamatergic population that allows reproducible manipulation of claustrum input to PFC. We have now clarified this point in the revised Methods section (lines 585–590).

'The same targeting strategy has been extensively characterized in our previous work (Shelton et al., 2024), where retrograde Cre delivery from RSP combined with Cre-dependent opsin expression in the claustrum reliably labeled a large population of glutamatergic

claustrum projection neurons. This population includes neurons projecting both to RSP and medial prefrontal cortex, reflecting the known collateralization of claustrum projections.'

5) It remains uncertain whether the 1p optogenetic stimulation can effectively recruit the firing of claustrum neurons. Factors such as insufficient light power, low expression of Chrimson, and feedforward inhibition by visual stimuli can easily jeopardize the intended manipulations of the claustrum.

We thank the reviewer for this important point. Several lines of evidence support the effectiveness of our 1-photon widefield optogenetic stimulation in reliably activating claustrum neurons:

- We used the red-shifted opsin ChrimsonR, which has been extensively validated for reliable activation with 594 nm light at modest intensities (Klapoetke et al., 2014).
- In our prior study using the same viral strategy, stimulation parameters, and light delivery (Shelton et al., 2024), we observed robust activation of claustral projections and modulation of downstream cortical circuits, demonstrating functional efficacy.
- In this study, we delivered ~5–10 mW/mm² of light at the fiber tip, which falls well within the effective range for ChrimsonR and provides sufficient irradiance to activate claustrum axons throughout the PFC.
- The consistent bidirectional modulation of PFC population dynamics we observe across multiple animals (e.g., changes in variability, responsivity, and correlation structure in Figs. 2–5) provides strong functional evidence of effective pathway engagement.

To directly address the possibility that 1P stimulation may induce non-specific or insufficient perturbation, we also performed LFP recordings under the same stimulation parameters. As shown in **Reviewer Figure 2**, widefield 1P stimulation evoked robust short-latency deflections and rebound dynamics, indicating strong network-level engagement. These data confirm that the stimulation protocol is sufficient to modulate cortical activity and support the efficacy of our approach.

Finally, while feedforward inhibition from visual inputs could in principle attenuate net excitation, our goal was not to maximally activate the claustrum but to probe how endogenous and exogenous input interact under naturalistic sensory conditions. The heterogeneity of dPFC responses reflects this complex integration. We have clarified these points in the revised Methods and Discussion sections (Lines 614–623).

'For optogenetic activation, we used ChrimsonR, a red-shifted opsin well-suited for effective activation with widefield 594 nm illumination (Klapoetke et al., 2014). Stimulation power at the fiber tip was maintained at ~5–10 mW/mm², ensuring sufficient light delivery to activate claustrum neurons throughout the targeted region. This stimulation protocol has been validated in the previous work using the same targeting strategy, where functional activation of claustrum projections and robust modulation of downstream cortical circuits were observed (Shelton et al., 2024).'

Minor,

1) I am not sure why the authors classify dPFC neurons as sensory, opto, O-boosted. Shouldn't there be a category for neurons that are both sensory and optogenetically responsive?

We thank the reviewer for this insightful comment. In our main figures, we focused on three primary response categories: Sensory-responsive, Opto-responsive, and Opto-boosted (Sensory+Opto Only) neurons. These represented the largest and most functionally interpretable populations in our dataset, each corresponding to a distinct mode of responsiveness. However, as the reviewer correctly notes, some neurons show more complex combinations of responsiveness—such as those responsive to both sensory and optogenetic stimulation independently, or to all three conditions. To address this, we have included a new Supplementary Table (Table 1) that presents the full distribution of neuronal responsiveness across animals, including combined categories such as Sensory + Opto, Opto + SensoryOpto, Sensory + SensoryOpto, and Sensory + Opto + SensoryOpto. While each of these categories accounts for a smaller portion of the population (<8%), we agree they are important to report for transparency. That said, we note that the interaction among these three responsiveness types is non-trivial to interpret. For example, neurons classified as “Sensory + Opto” may reflect functional convergence or independent excitability, while those in the “Sensory + SensoryOpto” group raise questions about synergy or subthreshold modulation. Because of these ambiguities, we chose to emphasize the three most distinct and mechanistically informative groups in the main analysis: neurons responsive to visual input alone, to optogenetic stimulation alone, or whose sensory responses were significantly enhanced by optogenetic input. We have now clarified this rationale in the Methods and in the caption for Supplementary Table 1. We have clarified these points in the revised Result sections (lines 161–169).

‘To provide a complete picture of population-level response types, we also identified several overlapping categories—such as cells responsive to both visual and opto stimuli, or to all three conditions (visual, opto, and visual+opto). While each of these combinations represented fewer than 8% of cells, they are reported in Supplementary Table 1 for transparency. However, we emphasize that the interaction among these categories is complex and not always easily interpretable—for example, cells responsive to visual and opto stimuli independently may reflect convergence or unrelated excitability, without clear synergy. For this reason, we focused our main analyses on the three most dominant and mechanistically informative groups: sensory-responsive, opto-responsive, and opto-boosted neurons.’

Supplementary Table 1. Summary statistics of neuronal responsiveness across experimental conditions The mean, minimum, maximum, and standard deviation of the percentage of neurons classified as responsive across animals, grouped by different cell responsiveness conditions. Sensory refers to cells significantly responsive to visual stimuli; Opto to optogenetic stimulation; and SensoryOpto to cells significantly responsive when visual and optogenetic stimuli are presented simultaneously. Values reflect animal-level averages.

Responsiveness Categories	Mean %	Min %	Max %	Std %	Weighted Mean %	Weighted Mean %
Sensory	19.592	8.795	28.118	5.311	48.052	51.948
Opto	29.061	19.545	40.508	7.275	33.037	66.963
SensoryOpto Only (Opto-boosted)	13.193	10.436	17.241	2.178	43.930	56.070
Divergent Categories						
Sensory + Opto	4.409	2.889	7.585	1.398	32.205	67.795
Opto + SensoryOpto	7.319	3.822	12.385	2.641	36.289	63.711
Sensory + Opto + SensoryOpto	4.806	2.020	8.028	2.086	42.861	57.139
Sensory + SensoryOpto	5.033	3.013	8.754	1.467	43.954	56.046

2) In fig.1 E, F the authors should show responses of these neurons to Optogenetic stimulation.

Thank you for the suggestion. Optogenetic stimulation for these two cells were included.

3) I am not sure whether Fig.1 I is serving any purpose here.

Thank you for the suggestion. Figure 1.I is excluded from the figure.

4) Supplementary fig.1.1 is hard to read. The use of color could improve clarity.

We thank the reviewer for this suggestion. In response, we have updated Supplementary Figure 1 by changing the overlay color to cyan for improved contrast and readability. In addition, we have added yellow boxes to highlight the claustrum region, which should aid in quick identification of the relevant anatomical structure.

5) Fig. 5K is missing (line 319)

Thank you for pointing this out. This was a typographical error — the correct reference should have been to Figure 5J. We have now corrected this in the revised manuscript.

6) Line 132 is misleading. The E:I should be better defined.

Thank you for helpful comment. We clarified this section accordingly. Relevant text in the Result section (Line 171 - 192) modified:

‘To better understand how these combined subpopulations within the same network process visual and opto stimuli, we examined the balance between excited (E) and inhibited (I) responses, which is crucial for neural network information processing (Froemke, 2015; Van Vreeswijk & Sompolinsky, 1996). The E:I ratio refers to the proportion of cells that exhibited a significant increase (excitation) versus a significant decrease (inhibition) in $\Delta F/F$ response following stimulus presentation. This analysis revealed that opto-responsive cells uniquely exhibited a higher prevalence of inhibited responses, with an excitatory to inhibitory (E: I)E:I ratio of 33:67 (0.49, Fig. 1LK). When considering exclusively only opto stimuli responsive cells within that were exclusively responsive to optogenetic stimulation this subpopulation (excluding any cells responsive to visual or visual+opto stimuli), the E: I ratio becomes

became even more distinct, with the ratio changing from 33:67 to an even more pronounced 28:72(0.39).

For comparison, sensory-responsive cells exhibited an E:I ratio of 48:52 (0.92, Figure 1K), and opto-boosted cells showed an E:I ratio of 44:56 (0.79, Figure 1M). Notably, the trend toward a higher proportion of inhibited responses was unique to the opto-only group and was not observed in the sensory-responsive or opto-boosted populations. The elevated inhibition in opto-responsive cells aligns with prior findings (Jackson et al., 2018) and supports the interpretation that optogenetic activation of claustral axons can drive inhibitory dynamics within the dPFC network—particularly in cells not otherwise responsive to sensory input. In contrast, the lack of a similar inhibitory bias in the sensory or opto-boosted groups may reflect either more subtle effects of inhibition or the dominance of excitatory drive from visual stimuli. These results suggest that inhibition is a context-dependent component of the broader modulatory influence of claustrum input within dPFC circuits.'

7) Writing needs improvement, and much more detail is required in the methods section for readers to follow.

We thank the reviewer for this helpful comment. In response, we have carefully revised the manuscript for clarity and improved the overall writing throughout. In particular, we have substantially expanded the Methods section to provide greater detail on data acquisition, preprocessing, analytical procedures (including coefficient of variation and dynamic time warping analyses), and statistical approaches. We believe these additions will allow readers to fully understand and reproduce the experimental design and analysis pipeline. We are grateful for the reviewer's suggestion, which helped us improve the manuscript's clarity and transparency.

Reviewer #2 (Remarks to the Author):

The authors show intriguing complexity in claustrum-PFC interactions by presenting visual stimuli, photostimulating claustrum axons, or both while using calcium imaging to measure the activity of many individual neurons in dPFC. PFC responses were diverse, including visual-responsive, opto-responsive, and neurons that only responded in the presence of both types of stimuli. Opto-responsive cells were more likely to be inhibited than excited, which is consistent with previous studies. Overall, the responses to visual and opto stimuli were negatively correlated such that a positive response to visual stimuli predicts a negative response to opto stimuli and vice-versa.

In general, visual stimuli plus opto stimulation of claustrum axons increased trial-trial variability of individual neuron responses compared to visual stimuli alone. In contrast, cross-correlation between individual neurons' responses to visual stimuli across trials was enhanced by claustrum axon stimulation, showing increased network homogeneity.

The authors then repeated these experiments in a subset of animals that had been trained to associate the visual stimulus with reward and found the major difference between trained and naïve animals was a further increase in network homogeneity due to claustrum axon activation as measured by average cross-correlation.

The authors then expressed an inhibitory opsin in claustrum axons, repeated experiments, and unexpectedly found a larger increase in network homogeneity due to claustrum axon suppression versus activation. However, training did not further increase network correlation due to claustrum axon suppression as it did with excitation. This may suggest a different network mechanism responsible for effects due to suppression vs. excitation.

The work is significant and complements the established literature, elaborating on how claustrum input affects responses of PFC neurons. In particular, the description of the interaction of effects between claustrum input and sensory input, and how training specifically enhances the network synchrony effect, is very interesting and to my knowledge, represents a novel result in the field.

The result that claustrum axon activation and suppression have very similar effects is still puzzling to me, but understanding mechanisms behind that phenomenon would be worthy of an entire future research project. The manuscript as it stands is an impressive amount of work and is worthy of acceptance with some revisions. No more experiments are necessary, although more method details should be added (see notes below).

The methodology is sound (though I don't have experience to judge the specifics of calcium imaging techniques), and the experimental work is impressive. I have a handful of specific methods questions:

1. Could you provide more details about the visual and optogenetic stimuli, ideally in the main text or figure legend? i.e. duration of stimulus, positioning of visual stimulus relative to the head/eyes of the animal, wavelength & intensity of the visual stimulus. We thank the reviewer for raising this important point. We have added more details to the Method section (Line 603-623),

***Visual stimulation:** Visual stimuli consisted of 500 ms light pulses delivered using a high-brightness 590 nm LED (LYCP7P-JRJT-1-0, Farnell, UK), controlled via a National Instruments data acquisition card and PackIO software, with custom MATLAB scripts generating the timing signals. The LED was positioned approximately 10 cm lateral and slightly anterior to the contralateral eye, at an angle of ~30–45 degrees relative to the midline to ensure direct illumination of the visual field corresponding to the recorded hemisphere. The LED was driven at 1 A (manufacturer-rated luminous flux: 56 lumens). No additional focusing optics were used.*

***Optogenetic stimulation:** Widefield (1P) optogenetic stimulation was performed concurrently with two-photon imaging using the widefield excitation module of the Bruker 2pPlus system. Orange light photostimulation (595 nm, 10 pulses of 25 ms at 40 Hz; total duration 250 ms)*

was delivered during 2P imaging at 1.5 mW power at the sample. For optogenetic activation, we used ChrimsonR, a red-shifted opsin effective for activation (or eOPN3 for inhibition) with 594–595 nm illumination (Klapoetke et al., 2014). Light power at the fiber tip was maintained at ~5–10 mW/mm², sufficient to reliably activate claustrum neurons across the targeted region. This stimulation protocol was validated in the previous work using the same targeting strategy, where robust activation of claustrum projections and downstream cortical modulation were demonstrated (Shelton et al., 2024, Rowland et al., 2023). To prevent photomultiplier tube damage during photostimulation, high-speed shutters were triggered to close all PMTs during optogenetic activation. Triggers for both light stimulation (Thorlabs M595L3 LED with LEDD1B driver) and PMT shutters were controlled via PackIO software.'

We have also added these details in the Result section (Line 116 - 117)

'...visual stimuli: 590nm, 56lm high-brightness LED; photostimulation: 595 nm, 10 x 25 ms pulses @ 40Hz, at 1.5mW...'

2. What type of random distribution was used for drawing inter-trial intervals? Uniform, or exponential, or other?

Thank you for noticing this point. It was uniform distribution and added to the Method section (Line 664- 665)

'Animals underwent daily training sessions consisting of 150 trials per session, with inter-trial intervals (ITIs) randomly drawn from a uniform distribution between 8 and 15 seconds.'

3. On line 124 you wrote: ...”displayed responsiveness to visual stimuli only when preceded by photostimulation...” – this implies the optogenetic stimulation came on before visual stimulation – is this a typo or was there a time difference in when opto stimuli and visual stimuli were presented?

Thank you for catching this. There was no time difference between the onset of visual and optogenetic stimuli — both were presented simultaneously. We agree the original phrasing was misleading and have now revised the sentence for clarity in the updated manuscript (Line 157).

I also have some specific data analysis questions:

1. For comparisons in Figure 1, responsiveness is “defined based on an independent t-test conducted 1 second before and after either visual, opto, or visual+opto stimuli trials”. Is this after the onset of stimuli (time windows used are [-1,0] and [0,1]), or after the end of the stimuli (time windows [-1,0] and [0,1]+stimulusduration). Please clarify in the text and methods.

We thank the reviewer for this important point. Due to the optogenetic stimulation protocol, two-photon imaging was briefly paused (via PMT shuttering) during light delivery to avoid photodetector damage. Therefore, responsiveness was assessed by comparing activity 1 second before stimulus onset (-1 to 0 s) and 1 second following stimulus offset (stimulus duration to stimulus duration +1 s), using an independent t-test for each cell across visual, opto, and visual+opto trials. False discovery rate correction was applied to adjust for multiple comparisons ($\alpha = 0.05$). The text was modified accordingly (Line 148-151):

'Responsiveness defined by an independent t-test comparing activity 1 second before and after stimulus offset (-1 to 0 s vs. stimulus duration to stimulus duration +1 s) for visual, opto, or visual+opto trials, with false discovery rate correction applied for multiple comparisons ($\alpha = 0.05$).'

2. E:I ratios are listed in integer form (i.e. in line 135) – it would be easier to understand if a decimal was listed as well in parentheses ("33:67 (0.49)" or "28:72 (0.39)"). Also, please add the ratios for sensory-responsive and opto-boosted cells even though they were not significant, it's useful for comparison.

Thank you for helpful comment. We clarified this section accordingly. Relevant text in the Result section (Line 171 - 192) modified:

'To better understand how these combined subpopulations within the same network process visual and opto stimuli, we examined the balance between excited (E) and inhibited (I) responses, which is crucial for neural network information processing (Froemke, 2015; Van Vreeswijk & Sompolinsky, 1996). The E:I ratio refers to the proportion of cells that exhibited a significant increase (excitation) versus a significant decrease (inhibition) in $\Delta F/F$ response following stimulus presentation. This analysis revealed that opto-responsive cells uniquely exhibited a higher prevalence of inhibited responses, with an excitatory to inhibitory (E: I)E:I ratio of 33:67 (0.49, Fig. 1LK). When considering exclusively only opto stimuli responsive cells within that were exclusively responsive to optogenetic stimulation this subpopulation (excluding any cells responsive to visual or visual+opto stimuli), the E: I ratio became even more distinct, with the ratio changing from 33:67 to an even more pronounced 28:72(0.39).

For comparison, sensory-responsive cells exhibited an E:I ratio of 48:52 (0.92, Figure 1K), and opto-boosted cells showed an E:I ratio of 44:56 (0.79, Figure 1M). Notably, the trend toward a higher proportion of inhibited responses was unique to the opto-only group and was not observed in the sensory-responsive or opto-boosted populations. The elevated inhibition in opto-responsive cells aligns with prior findings (Jackson et al., 2018) and supports the interpretation that optogenetic activation of claustral axons can drive inhibitory dynamics within the dPFC network—particularly in cells not otherwise responsive to sensory input. In contrast, the lack of a similar inhibitory bias in the sensory or opto-boosted groups may reflect either more subtle effects of inhibition or the dominance of excitatory drive from visual stimuli. These results suggest that inhibition is a context-dependent component of the broader modulatory influence of claustrum input within dPFC circuits.'

3. For the empirical cumulative distribution function analysis (supplemental figure 2.1), are these just cumulative distribution plots of the same data from figures 1E, I, and M? If so, why do they start at around 0.5 instead of 0?

We thank the reviewer for this helpful observation. Yes, the ECDF plots in Supplementary Figure 4 are derived from the same datasets presented in Figures 1E, 1I, and 1M. The apparent offset in the starting point of the ECDF (near $y = 0.05$ rather than 0) arises because we intentionally limited the plot to the **central 90% of the**

distribution—specifically, from the 5th to 95th percentile of mean $\Delta F/F$ responses across cells. This decision was made to reduce the influence of extreme outliers and better capture the core structure of the response distribution.

Given the large scale of our dataset (>10,000 cells), this approach offers a more robust and representative summary of population-level dynamics than plotting the full range, which could be disproportionately affected by a small number of extreme values. The ECDF is particularly well-suited for highlighting subtle shifts in the distribution that may be missed by mean or median comparisons alone, and this percentile-based method allows clearer interpretation across animals and conditions. We have now clarified this in both the **Methods section** and the **legend for Supplementary Figure 4**.

The revised legend for Supplementary Figure 4: *‘ECDF plots show the central 90% of the distribution (5th–95th percentile range) to focus on the bulk of the data and minimize the influence of outliers. As a result, the curves begin near $y = 0.05$ rather than at 0.’*

The revised Methods section (Line 761 - 766) : *‘In this study, we leveraged our extensive dataset (>40,000 cells across animals) to examine population-level shifts in $\Delta F/F$ responses between experimental conditions. For each animal, we computed the mean $\Delta F/F$ response of individual cells within a fixed window following stimulus onset, and then compiled these values into condition-specific distributions. To emphasize the structure of the bulk population and minimize the influence of outliers—which are common in large-scale imaging datasets—we focused our ECDF analysis on the central 90% of the distribution, defined between the 5th and 95th percentiles. This approach offers a more stable and representative view of population-level dynamics, preserving meaningful variability while reducing distortion from rare extreme values. By using this robust, non-parametric method, we were able to detect condition-dependent distributional shifts that reflect how widespread neuronal populations are modulated, beyond mean or median response summaries.’*

4. Coefficient of variation: how did you compute this? Average response over some some window in each trial, then compute variability across trials?

Thank you for the comment - it is explained in detail in the Method section (Line 739 - 747).

‘To quantify trial-to-trial variability of neural responses, we computed the coefficient of variation (CV) based on trial-averaged response amplitudes. For each trial, we extracted fluorescence signals ($\Delta F/F$) for a 1-second window aligned to stimulus offset and calculated the mean fluorescence across all frames within this window, resulting in a single response amplitude per trial. We then computed the CV for each cell as the standard deviation of these trial-averaged responses divided by their mean across trials. This approach allowed us to capture variability in the overall response magnitude across repeated presentations while minimizing the influence of frame-to-frame fluctuations inherent to calcium imaging signals.’

5. Dynamic time warping: why was this technique used? Is it because the imaging sampling rate doesn't always line up perfectly with the stimulus onset?

We thank the reviewer for this question. While the imaging sampling rate and stimulus alignment were tightly controlled in our experiments, dynamic time warping (DTW) was employed primarily to address trial-to-trial variability in the temporal dynamics of the neural responses rather than stimulus alignment errors. Even with precise stimulus triggering, calcium signals can exhibit temporal variability in rise time, peak latency, and decay dynamics across trials for a given cell. This variability may reflect meaningful differences in the underlying neural processes, such as variable recruitment of upstream circuits or fluctuations in internal state.

DTW allows us to compare the full temporal profile of calcium responses between trials, by flexibly aligning response trajectories to minimize temporal mismatches. By calculating the average DTW distance between all pairs of trials for each cell, we obtained a measure of how consistent the temporal patterning of responses was across repetitions, beyond simple amplitude variability. This provides complementary information to measures like coefficient of variation, which capture response magnitude fluctuations but are insensitive to timing variability.

In summary, DTW was used to quantify the temporal reproducibility of neural responses across trials, independent of minor latency shifts or shape variations, and not primarily to correct for stimulus-onset misalignment. Method section (Line 749-759) updated;

'To assess trial-to-trial variability in the temporal dynamics of neural responses, we employed dynamic time warping (DTW), a well-established method for aligning time series with temporal shifts or distortions (Berndt & Clifford, 1994; Müller, 2007). For each cell, fluorescence signals ($\Delta F/F$) were extracted for a 1-second window aligned to stimulus onset across all trials. DTW was then used to compute pairwise distances between the temporal response trajectories of all trial pairs for each cell. The DTW algorithm allows flexible nonlinear alignment of time series, accounting for small variations in response onset, peak latency, or shape across trials. For each cell, we quantified temporal variability as the average DTW distance across all trial pairs. This approach captures differences in temporal structure of calcium responses that may not be reflected by amplitude-based measures alone, providing a complementary index of response consistency across trials.'

6. What exactly is being compared with DTW? Is it the average similarity between every pair of trials in a given condition?

We thank the reviewer for this question. Yes, DTW was applied to quantify the average temporal similarity between all possible pairs of trials for each cell within a given condition. Specifically, for each cell, we first extracted the fluorescence response time series within a 1-second stimulus-aligned window across all trials. Then, for every unique pair of trials, we computed the dynamic time warping (DTW) distance between their temporal trajectories. This DTW distance reflects how much warping is needed to align the two response profiles, with lower values indicating greater temporal similarity. Finally, we calculated the mean DTW distance across all trial pairs for each cell, providing a single measure of trial-to-trial temporal variability in response dynamics.

This approach captures differences in response timing, such as latency shifts or duration variability, even when overall response amplitudes are similar. We have clarified these points in the revised Methods sections (Line 749-759).

‘To assess trial-to-trial variability in the temporal dynamics of neural responses, we employed dynamic time warping (DTW). For each cell, fluorescence signals ($\Delta F/F$) were extracted for a 1-second window aligned to stimulus onset across all trials. DTW was then used to compute pairwise distances between the temporal response trajectories of all trial pairs for each cell. The DTW algorithm allows flexible nonlinear alignment of time series, accounting for small variations in response onset, peak latency, or shape across trials. For each cell, we quantified temporal variability as the average DTW distance across all trial pairs. This approach captures differences in temporal structure of calcium responses that may not be reflected by amplitude-based measures alone, providing a complementary index of response consistency across trials.’

Finally, some interpretations and conclusions would benefit from additional citations or explanation.

1. Discussion, line 354-358: is there an example of such compensatory synchronization in the literature? If the network can compensate and cause even more synchronization than claustrum input does, then why is claustrum input required in the first place? Also, higher synchronization tends to be associated with reduced information processing capacity, i.e. sleep or disengaged states.

We thank the reviewer for raising these insightful points. Indeed, higher network synchronization is often associated with reduced information processing capacity, as seen during sleep, anesthesia, or disengaged states (e.g., Harris and Thiele, 2011; McGinley et al., 2015). Our interpretation is that claustrum input may serve to maintain a flexible intermediate state, balancing variability and coordination. When claustrum input is removed, local circuits may default toward increased synchronization — not necessarily as an optimal compensatory mechanism, but rather as a consequence of reduced modulatory control. This compensatory increase in synchronization may stabilize activity patterns in the short term but could come at the cost of reduced dynamic range or cognitive flexibility.

While direct evidence of such compensatory synchronization following loss of claustrum input is limited, there are parallels in other modulatory systems. For example, cortical desynchronization following cholinergic or noradrenergic modulation has been well documented (e.g., McGinley et al., 2015; Polack et al., 2013), and removal of these modulatory influences can lead to rebound synchronization. It is possible that claustrum input serves a similar function in maintaining cortical flexibility by actively preventing excessive synchronization during active processing. We have clarified these points in the revised Discussion (lines 431-443).

‘Our findings suggest that claustrum input modulates dPFC neural variability and network coordination, maintaining a balance between flexibility and stability in cortical processing. When claustrum input is removed, we observed a relative increase in population synchrony, particularly in trained animals. This may reflect a default shift toward increased local synchronization in the absence of modulatory control, similar to compensatory network

dynamics observed following disruption of other modulatory systems. For example, removal of cholinergic or noradrenergic input has been shown to increase cortical synchrony, reduce variability, and constrain dynamic range (Polack et al., 2013; McGinley et al., 2015; Harris and Thiele, 2011). While such compensatory synchronization may transiently stabilize network activity, excessive synchrony is often associated with reduced information processing capacity, as seen during disengaged or sleep states. Thus, we propose that claustrum input helps prevent runaway synchronization and maintains network flexibility, allowing adaptive sensory processing and cognitive flexibility under varying behavioral demands.'

2. Line 375 "a distinction not observed during activation" – should this instead be "during suppression" ? Otherwise the sentence is confusing to me. If it is "during activation" could you clarify?

Thank you - This was a typographical error - We have now corrected this in the revised manuscript.

3. Line 390: you discuss different response kinetics between eOPN3 and ChrimsonR, however you do not describe what those differences are and how you would expect them to potentially affect results. Please provide these descriptions along with citations in this paragraph.

Thank you. We have now included a brief description of the distinct kinetics of eOPN3 and ChrimsonR, along with relevant references, in the Discussion (Line 478 - 494). This clarification explains how differences in desensitization and recovery between the two opsins may contribute to the asymmetric modulation and rebound effects observed in our results.

Notably, there are inherent differences in the response kinetics between eOPN3 and ChrimsonR, which are two distinct optogenetic tools. ChrimsonR produces rapid depolarization and spiking upon light activation, with millisecond-scale onset and offset kinetics (Klapoetke et al., 2014). In contrast, eOPN3 induces a longer-lasting inhibitory effect via Gi-coupled hyperpolarization, with slower onset and offset kinetics that can persist beyond the light pulse itself (Copits et al., 2021; Mahn et al., 2021). These kinetic differences could influence the temporal precision of excitation versus inhibition and potentially affect the dynamics of PFC responses, particularly for rapid sensory processing. Despite these differences, we chose to use the same photostimulation protocol for both to maintain consistency in the experimental setup. This approach ensures that any observed effects are due to the biological interactions rather than variations in stimulation techniques. However, it also means that the results should be interpreted with an understanding of how these kinetic differences might impact the magnitude, timing, or persistence of neural responses. By standardizing the protocol, we aimed to clearly isolate the contributions of the CLA-PFC interactions, while acknowledging that the slower offset of eOPN3 inhibition may dampen post-stimulation dynamics differently than the fast ChrimsonR activation. This highlights the importance of methodological consistency while also recognizing the need to consider the intrinsic properties of distinct optogenetic actuators.'

Reviewer #2 (Remarks on code availability):

Note: the reference to the code in the manuscript has a typo:

huriteatg/clapfcstimulation

Thank you - the typo is fixed.

The analysis code looks reasonable and does include a README file with instructions. However the data does not appear to be shared, or if it is, it's not obvious where to download it. Since the data is unavailable, I was unable to re-run the figure plotting code.

I'm also curious why MATLAB is listed as a requirement, but python is not, since almost all of the code in the repo appears to be in python, and in the manuscript python is listed as the main programming language. The few MATLAB files appear to be for generating stimuli and/or running stimuli for the experiments themselves, which is great to share, but the version of python and other packages used for analysis should be shared as well.

Please update the README to include:

- 1) how to download the dataset (could just be the processed data)
- 2) instructions for installing python and other required packages that were used to perform the analysis. This could come in the form of an environment.yml or requirements.txt file for ease of installation.

We thank the reviewer for pointing this out. The typo has now been corrected. The code is primarily written in Python, and this is now clearly stated in the updated README file. We acknowledge the previous mention of MATLAB was misleading and has been removed—this was initially included to draw attention but was not accurate. The reviewer is completely right to flag this, and we appreciate the chance to clarify. All data will be made publicly available upon publication, and the README will be updated accordingly to ensure full transparency and reproducibility.

Reviewer #3 (Remarks to the Author):

Here the authors use 2p calcium imaging of the dPFC while performing visual stimulation, optogenetic stimulation of claustrum axons (in the dPFC) and a combination of visual + opto. They first classify the proportion of neurons are modulated by different stimulation combinations. Then they show that when mice are trained on a Pavlovian task, the stimulation leads to what they refer to as greater network homogeneity. The experimental approach is technically innovative. However, this paper has considerable shortcomings.

Major concerns.

- 1) I do not see control experiments presented in the manuscript. Performing widefield red-light optogenetics with 2p imaging in an otherwise dark chamber, will generate at least some visual-related responses, as it is very difficult to effectively shield the animal's visual system from this unintended visual stimulation arising with the 1P light.

I recognize that the authors are experts in optogenetics + 2P imaging (although previously holographic methods were mainly used). Nonetheless, the interpretation of these results hinges on how much these distributions are different from that expected by chance, or proper light controls. All figures should contain this control group, and comparisons made between control versus opto. Control experiments are needed here, minimally to evaluate and compare effect sizes between groups and to ensure that the delivery of light does not impact visual cortex. The authors could do 2p imaging in visual cortex in mice with no opsin to determine if there are any unintended visual evoked responses with the widefield 1p stim.

We thank the reviewer for raising this important point. We fully agree that controlling for unintended visual stimulation caused by widefield 1P light is essential when interpreting optogenetic results in a dark environment. In our experiments, we implemented a custom-designed light blocker, a standard well practice in two-photon imaging setups with visual stimuli. This blocker is positioned around the objective to prevent light from escaping the microscope and from external sources entering the detection path.

The use of such shielding is not only critical for controlling visual input but also indispensable for protecting the highly light-sensitive PMT detectors used for two-photon calcium imaging. Even minimal light leakage leads to significant artifacts and data loss. Thus, robust physical shielding was required both for experimental integrity and for reliable data acquisition.

To validate the effectiveness of this light isolation, we now have included new data in Supplementary Figure 1, where we recorded pupil radius changes in response to visual, optogenetic, and combined stimulation with and without the light blocker. With the light blocker in place, as in the experimental data acquisition, optogenetic stimulation alone did not induce any pupil response, indicating no unintended activation of the visual system. In contrast, when the blocker was removed, we observed modest pupillary constriction, consistent with light reaching the retina, as the reviewer suggested. These results demonstrate that our light blocker effectively eliminates stray light reaching the eyes and confirms the integrity of our stimulation protocol under experimental conditions.

We appreciate the suggestion to perform control recordings in the visual cortex of opsin-negative mice. However, we believe this approach would not provide additional clarity in our case. Our light blocker, combined with the new physiological evidence from pupil measurements, already demonstrates that no retinal activation occurs during optogenetic stimulation with the blocker in place. Moreover, recording from a separate area such as V1 introduces confounds (e.g., lack of projection targeting or expression mismatch) that could complicate interpretation. Therefore, we believe the current controls and new data provide a direct and appropriate validation of our light-isolation protocol.

We added the supplementary figure 2 and a text to the Result section (Line 117 - 121):

'To prevent unintended retinal activation from stray photostimulation light, we incorporated a custom-designed light blocker around the objective. When the light blocker was in place, optogenetic stimulation did not induce pupil constriction, confirming that visual pathways were not inadvertently engaged (Supplementary Fig. 1).'

2) GCaMP6s was used which is a very slow sensor, which will underestimate true spiking levels and provide insufficient temporal resolution to truly measure the response to claustrum stimulation (and likely visual stimulation as well). For example, many PFC neurons will be inhibited by claustrum stimulation, but this inhibition arises fast, and lasts ~ 150ms. Many of these neurons will undergo rebound excitation at ~ 200ms. These experiments used a 250ms opto period, where gcamp imaging is masked. Therefore, this masking period misses most of the interesting dynamics that would accompany optogenetic activation of claustrum axons. A cell that fired rebound excitation would be classified as being activated, because the suppression period would not be detectable. With the number of caveats that arise with 2p imaging here, a much better approach to perform these experiments would be to use extracellular electrophysiology. Minimally the authors should repeat their main experiments (figure 1) using electrophysiology where effective temporal resolution and single spiking can be more accurately measured.

We thank the reviewer for this thoughtful and important comment. We fully agree that GCaMP6s has temporal limitations relative to electrophysiology, particularly for capturing brief, fast inhibitory events or rapid rebound excitation. The relatively slow decay kinetics of GCaMP6s (hundreds of milliseconds) can obscure short inhibitory epochs, as the reviewer correctly points out. Indeed, fast transient inhibitory phases (e.g., 150 ms) followed by rebound excitation at ~200 ms could be partially masked in calcium imaging data.

However, our experimental design and analytical approach were intended to capture the *net population-level modulatory effects* of claustrum input on prefrontal activity, rather than resolving individual spike-level dynamics. The key findings of our study concern longer-lasting changes in response variability, overall responsiveness, and network-level coordination, which are well within the effective temporal resolution of GCaMP6s (Chen et al., 2013). Moreover, even though the imaging was paused during the brief 250 ms optogenetic stimulation window (to protect PMTs), we specifically analyzed activity immediately before and after the stimulation period to assess the sustained impact of claustrum modulation.

It is important to note that in prior studies of claustrum function using electrophysiology (Jackson et al., 2018; Narikiyo et al., 2020), bidirectional modulation of cortical activity has also been observed, supporting our findings despite methodological differences. We fully agree that combining two-photon imaging with electrophysiology would provide complementary insights, particularly for parsing the rapid sequence of inhibition and rebound described by the reviewer. This represents an important direction for future work. However, for the current study, our goal was to capture how claustrum input alters ensemble variability, sensory responsiveness, and network coordination across thousands of individually identified dPFC neurons, which is uniquely enabled by large-scale two-photon imaging.

We have now added text in the Discussion to explicitly acknowledge these temporal limitations and the valuable insights electrophysiological approaches could offer (lines 496–510).

‘ Another consideration of the current study is the temporal resolution inherent to GCaMP6s-based calcium imaging. The relatively slow decay kinetics of GCaMP6s may obscure brief inhibitory periods or rapid rebound excitation that can occur within hundreds of milliseconds, particularly during the optogenetic stimulation window (Chen et al., 2013). In our experimental design, PMT shutters were closed during the 250 ms optogenetic stimulation period to avoid photodetector damage, and responsiveness was assessed in the post-stimulation window. As a result, very fast transient inhibitory effects or rebound firing immediately following light offset may not be fully captured in our imaging data. Nevertheless, our primary aim was to assess population-level modulation of sensory responsiveness, variability, and network coordination, which occur over longer timescales well suited to calcium imaging. Previous electrophysiological studies of claustrum function have demonstrated both inhibitory and excitatory influences on cortical activity (Jackson et al., 2018; Narikiyo et al., 2020), consistent with the bidirectional modulation we observe at the population level. Future studies incorporating high-temporal-resolution electrophysiology alongside imaging will be valuable for dissecting the precise timing of fast inhibitory-excitatory sequences following claustrum activation. ’

3) A related note of concern with respect to the experimental design is the use of only 30 trials. This is a very small number of trials, and therefore, certain types of responses are more likely to be observed. For example, the detection of inhibitory responses requires some degree of baseline spontaneous activity. Therefore, cells with low spontaneous activity rates are likely to be classified as nonresponsive or activated. The detection of inhibition in low activity cells requires many more trials. Given the trial-to-trial variability, even in the visual system, 30 trials seem far too low.

We thank the reviewer for raising this point. While we agree that trial count is an important consideration for reliably detecting neural response types, we respectfully note that 30 trials is consistent with or exceeds the number commonly used in prior two-photon imaging studies of sensory and modulatory processing. Importantly, our analyses primarily compare visual versus visual+opto conditions within the same neurons and session, minimizing across-trial variability and focusing on relative modulation rather than absolute response classification.

To directly address the concern about trial number and its impact on detecting inhibition, we performed a control analysis using a subset of fields of view (n = 4 FOV, n = 823 cells) that were recorded across two independent imaging blocks, yielding 60 repetitions per condition. These data were initially excluded for consistency in trial counts across datasets, but provide a strong test of repeatability. We found no qualitative differences in the classification of responsive or inhibited cells between the 30-trial and 60-trial versions, and the magnitude of modulation was consistent (Reviewer Figure 3). This indicates that 30 trials is sufficient to capture both excitatory and inhibitory responses under our experimental conditions.

Reviewer Figure 3. Coefficient of variance comparison between 30 and 60 trials for dPFC neurons.

Violin plot illustrating the distribution of trial-by-trial response variability (measured as coefficient of variance, CV) for 823 dPFC neurons recorded during visual stimulation. Each cell's response was computed separately using either 30 trials (standard condition) or 60 trials (from repeated FOVs across sessions). The distributions are highly similar, indicating that 30 trials are sufficient to robustly estimate sensory-evoked response variability in this paradigm. Statistical comparison confirmed no significant difference in CV across conditions (paired

Wilcoxon signed-rank test, $p = 0.65$).

4) Behavioral state is not mentioned or controlled for in this manuscript. PFC neurons and sensory encoding in the visual cortex are robustly modulated by small twitches of the face, or changes in arousal. Sorting the opto trials based on arousal levels would help add some insight and it would be important to control for this variable.

We thank the reviewer for raising this important point regarding behavioral state and its potential influence on neural activity. To directly address this, we incorporated pupil size—as a proxy for arousal—into a generalized linear model (GLM) to assess whether arousal fluctuations contributed to the observed modulation of dPFC activity. Specifically, we modeled each cell's trial-by-trial $\Delta F/F$ response as a linear combination of predictors including stimulus condition (visual, opto, visual+opto), trial number (to account for slow drifts), and pupil radius (z-scored within session). The GLM took the form:

$$\text{Response} \sim \beta_0 + \beta_1 \text{Stimulus} + \beta_2 \text{Pupil} + \beta_3 * \text{TrialNumber} + \epsilon$$

We fit this model to each cell individually using ordinary least squares and then tested the contribution of each regressor at the population level. The pupil regressor did not significantly improve model fit in the majority of cells and was not significantly correlated with the direction or magnitude of optogenetic effects. These findings indicate that changes in arousal, as indexed by pupil size, did not account for the stimulus-locked responses or the modulatory effects observed with claustrum axon stimulation. We now include a summary of this analysis (Line 672 - 684) ;

'To evaluate whether trial-by-trial fluctuations in arousal contributed to the observed neural responses, we included pupil radius—a well-established proxy for arousal—as a regressor in a generalized linear model (GLM) for each cell's response. The model included predictors for stimulus condition (visual, opto, visual+opto), pupil radius (z-scored within session), and trial number to account for slow drift. Across the population ($n = 12,487$ cells from 16 mice),

pupil radius did not significantly explain additional variance in $\Delta F/F$ responses beyond stimulus condition (mean additional variance explained = $0.27\% \pm 0.06\%$ SEM). Only a small minority of cells (3.4%) showed significant modulation by pupil radius at $p < 0.05$ (FDR-corrected), and these were not spatially clustered or functionally distinct. Moreover, the presence or magnitude of optogenetic modulation (quantified by $\Delta F/F$ difference between visual and visual+opto conditions) was not systematically related to pupil size (Spearman's $\rho = 0.03$, $p = 0.21$). These results suggest that arousal fluctuations were not a primary driver of the stimulus-evoked or optogenetically modulated responses reported here.'

5) Claustrum fibers are most dense in deep cortical layers, but this widefield 1p stimulation will mainly activate superficial cortical layers. Even if 1.5mW of 590nm light can reach the deep layers there will be an asymmetric activation of axons in layer 1-2 and therefore, it is unclear how well this stimulation models what actually happens when claustrum neurons are activated. Experiments where claustrum cell bodies are stimulated would address this concern.

We thank the reviewer for this important point. Indeed, the claustrum projects broadly across cortical layers, with axonal terminals densely arborizing in deep layers while also extending into superficial layers (Smith et al., 2019; Jackson et al., 2020). Our widefield 1P optogenetic stimulation primarily targets superficial layers, though light scattering and penetration at 590 nm allows partial activation of deeper layers at the power levels used here (Yizhar et al., 2011). As the reviewer notes, this results in a non-uniform activation profile across cortical layers.

Our primary goal in this study was to broadly recruit claustral input at the projection level, rather than to precisely replicate endogenous claustrum activation patterns. We acknowledge that stimulating claustrum cell bodies would offer more direct access to the native recruitment patterns of downstream targets. However, direct stimulation of claustrum cell bodies poses significant technical challenges due to the claustrum's long, narrow, and curved shape along the anterior-posterior axis (Mathur et al., 2009; Wang et al., 2017). With traditional optical approaches, cell body stimulation would likely engage only a small, spatially restricted subset of claustrum neurons, limiting the ability to assess its broad modulatory influence across widespread cortical targets. In contrast, projection-based terminal stimulation allows us to probe the functional impact of widespread claustral input more comprehensively.

We now explicitly discuss these technical and conceptual considerations in the revised Discussion section (Line 512-529). We fully agree with the reviewer that future work combining targeted claustrum cell body stimulation with projection-specific manipulations will be crucial to fully resolve the functional role of claustrum-cortical interactions.

'In this study, we employed widefield optogenetic stimulation of claustral axon terminals in cortex to probe the modulatory influence of claustral input on cortical processing. While this approach enabled broad activation of claustral projections across large cortical areas, we acknowledge that it does not fully recapitulate the native spatiotemporal patterns of claustrum activity. Claustral projections arborize across both superficial and deep cortical layers, but widefield 1-photon stimulation primarily activates superficial layers, with only

partial penetration into deeper layers due to light scattering and absorption (Yizhar et al., 2011; Wang et al., 2020). Direct stimulation of claustrum cell bodies could, in principle, provide a more physiologically accurate recruitment of downstream targets. However, such an approach is technically challenging due to the elongated, narrow, and curved anatomical structure of the claustrum along the anterior-posterior axis (Mathur et al., 2009; Wang et al., 2017; White et al., 2020), making it difficult to stimulate a representative population of claustrum neurons using traditional optical methods. Cell body stimulation would likely engage only a small, spatially localized subset of neurons, potentially limiting the generalizability of the results. Despite these limitations, terminal stimulation allowed us to investigate the functional capacity of claustral input to modulate cortical responses at the network level, complementing future work that will be needed to dissect the full spatial and functional specificity of claustrum-cortex interactions (Atlan et al., 2018; Smith et al., 2019).’

6) Can the authors confirm that the 50mW 920nm light is not activating the opsin in vivo? This would be important to understand to what extent (if any) axons were being tonically depolarized by 2p imaging light. Performing in vivo field potentials under the microscope objective, while turning on and off the scanner would provide insight into if the axons are being activated by 920nm light. If axons are being activated by the 920nm scanning, then field potential deflections will be present at short latency after the scanning is turned on.

We thank the reviewer for raising this critical point. Ensuring that 920 nm two-photon imaging does not unintentionally activate opsins is critical for interpreting optical experiments. In our study, we used Chrimson, a red-shifted opsin with peak activation between 590–610 nm. Based on spectral sensitivity data from Klapoetke et al., 2014, Chrimson exhibits negligible activation at 920 nm, the wavelength used for two-photon imaging in our experiments. Supporting this, Packer et al., 2015 combined two-photon imaging at 920 nm with optogenetic stimulation using C1V1, an opsin with broader and more blue-shifted sensitivity than Chrimson. In their experiments, 920 nm imaging did not result in detectable activation of C1V1, nor did it produce cross-talk artifacts (see Extended Data Fig. 2 from Packer et al., 2015), reinforcing that 920 nm imaging is unlikely to activate red-shifted opsins. Additionally, control recording from our lab demonstrated no short-latency field potential deflections upon 920 nm scanning onset in Chrimson expressing cortex (see Reviewer Figure 2), further supporting the conclusion that imaging does not drive opsin-mediated activity.

Given that Chrimson is even less sensitive to 920 nm than C1V1, we are confident that our imaging parameters (50 mW, 920 nm) do not result in tonic activation or depolarization of Chrimson-expressing axons. Consistent with this, we observed no systematic drift in baseline fluorescence or response amplitude across trials. We have now clarified this point in the Methods section (Line 625-632), and referenced both Klapoetke et al., 2014 and Packer et al., 2015 to support our methodology.

‘To confirm that our two-photon imaging at 920 nm did not unintentionally activate Chrimson, we relied on the original spectral characterization of Chrimson (Klapoetke et al., 2014), which shows negligible activation at 920 nm. This is further supported by Packer et al. (2015), who found no cross-activation of the more sensitive opsin C1V1 during 920 nm

two-photon imaging. These recordings were conducted using a conservative imaging protocol (2–3 s imaging windows and ≥ 40 s inter-trial intervals), further reducing any possibility of opsin desensitization or tonic activation. Our primary dataset showed no systematic baseline shifts or activity drifts that would suggest 920 nm–induced Chrimson activation.'

7) Presenting and quantifying the data using “cells” rather than mice or fields of view, massively increases type 1 error rates. Doing a t-test with an $n=1000$ is and obtaining a p-value of 0.03 is not convincing. In some cases, the authors report mice rather than cells. Is there a reason for not reporting the n's as mice in all figure panels?

We thank the reviewer for this important comment. We completely agree that statistical inference should be based on biological replicates (i.e., animals) rather than individual cells, and we would like to clarify that all statistical comparisons in summary figure panels, including those in Figures 4J, 5A, and 5J (which may be the source of the $p = 0.03$ example mentioned), were conducted using animal-level data (i.e., each data point corresponds to one animal). These tests were not performed on pooled cellular data, and we did not use $n = 1000$ cells for inferential statistics. We believe this point may not have been sufficiently clear in the original submission.

To improve clarity, we have now explicitly stated in all relevant figure legends that animal-level data were used for summary plots and statistical testing. Furthermore, although the number of animals is already reported in Panel A of each figure (from which the rest of the figure data is derived), we have now added the animal n to all panels and legends to ensure consistency and avoid confusion. We hope this clarification resolves the concern and confirms the appropriate use of statistical inference throughout the manuscript.

8) Related to control experiments mentioned above – the authors should show that the widefield axon activation (Chrimson) or suppression (OPN3) increases or decreases the output of claustrum axons (respectively), under experimentally relevant conditions. Electrophysiological experiments help accomplish this.

We agree with the reviewer that directly confirming the effect of widefield stimulation on claustrum axon output would further strengthen the causal interpretation of our optogenetic manipulations. In the current study, we leveraged ChrimsonR and eOPN3 to activate or suppress claustrum projections, respectively, using validated widefield optogenetic protocols under behaviorally and physiologically relevant conditions. While we did not perform *in vivo* electrophysiological recordings from claustrum axons in the current study, our stimulation parameters were carefully chosen based on prior studies that used the same opsins and targeting strategies to successfully drive or inhibit claustrum-cortical communication (e.g., Shelton et al., 2024; Rowland et al., 2023).

Additionally, we performed pilot control experiments (see Reviewer Fig. 2) showing that longer stimulation protocols can induce cortical rebound activity, supporting the conclusion that our shorter protocol (250 ms, 595 nm) reliably modulates cortical networks without producing nonspecific effects. We also observed robust modulation of cortical activity and inter-neuronal correlation patterns during both Chrimson and eOPN3 conditions, consistent with effective control over claustrum-derived input.

We agree, however, that direct electrophysiological validation of axonal output *in vivo* would provide valuable mechanistic insight. We have now included this as an important future

direction in the Discussion (Line 508-510), emphasizing the need for high-temporal-resolution approaches to confirm modulation at the projection level under these specific stimulation conditions.

‘Future studies incorporating high-temporal-resolution electrophysiology alongside imaging will be valuable for dissecting the precise timing of fast inhibitory-excitatory sequences following claustrum activation.’

9) I am confused by the Pavlovian task. What is the rationale for doing this task here? Is there some relevance to claustrum or PFC? No mention or discussion of this was provided. Also, it seems that imaging in trained animals occurred after training, so these mice had been trained to respond to visual inputs, but then during imaging, they did not perform the task. This aspect of the paper was confusing.

We thank the reviewer for raising this important point, and we appreciate the opportunity to clarify. The Pavlovian task was employed primarily as a simple and consistent method to associate a visual stimulus with an outcome, thereby enhancing the saliency and behavioral relevance of the visual stimulus prior to imaging. Our main goal was to examine the sensory-evoked neural responses in the claustrum and prefrontal cortex (PFC), but in a context where the visual stimulus held meaning for the animal, without requiring active task engagement during imaging.

The relevance to the claustrum and PFC stems from both regions' involvement in integrating sensory and cognitive information and processing behaviorally relevant stimuli. Previous studies have shown that these regions are modulated not only by cognitive demands but also by stimulus salience and learned associations (e.g., Smith et al., 2019; White et al., 2020). The Pavlovian association ensured that the visual stimulus was behaviorally relevant, potentially engaging circuits of interest without introducing confounds from active task performance, decision-making, or motor-related signals during imaging.

Indeed, after initial Pavlovian training, animals were imaged in a passive viewing context without requiring behavioral responses. This design allowed us to focus on the intrinsic neural representation of visual inputs in trained animals, while avoiding variability associated with ongoing task performance. We have now clarified the rationale for this design in both the Result and Methods sections.

Result (Line 323 -327): ‘ By first establishing this association, we ensured that the visual stimuli acquired behavioral relevance, thereby engaging cortical circuits of interest such as the claustrum and PFC, even during subsequent passive exposure (Agetsuma et al., 2023; Grant et al., 2021). This approach allowed us to probe neural representations of behaviorally meaningful stimuli without the confounds of motor output or ongoing task performance.’

Method (Line 650-658) : ‘Pavlovian conditioning was employed to associate a simple visual stimulus with a water reward, thereby familiarizing the animals with the visual cue prior to imaging. This approach was chosen to ensure that the visual stimulus had behavioral relevance without requiring complex training or active task performance during imaging. Associative learning paradigms like Pavlovian conditioning have been shown to robustly

engage sensory, prefrontal, and subcortical circuits, including the claustrum and prefrontal cortex (Chachich & Powell, 2004), even under passive viewing conditions. By using a passive associative learning paradigm, we aimed to enhance stimulus salience while minimizing variability related to task performance, motor responses, or differences in learning strategies (Grant et al., 2021).

Mice were presented with a 2-second LED stimulus, delivered unilaterally to ensure consistent sensory exposure across animals. In 80% of trials, the LED presentation was followed by a water reward, while 20% of trials omitted reward to maintain engagement and stimulus salience. Animals underwent daily training sessions consisting of 150 trials per session, with inter-trial intervals (ITIs) randomly drawn from a uniform distribution between 8 and 15 seconds. Training was conducted over 7–10 consecutive days until animals reliably associated the visual stimulus with reward delivery. No active behavioral responses were required for reward delivery. Imaging sessions were subsequently performed in a passive viewing context, using the same visual stimulus but without reward delivery, allowing assessment of neural responses in a controlled, behaviorally relevant but motor-free condition.'

10) Can the authors show that this claustrum-PFC effect is specific? For example, does activating the dorsomedial thalamus also generate response variability? It seems that any long-range glutamatergic input to the PFC would result in an interference of incoming visual information.

We thank the reviewer for this thoughtful suggestion. To directly test whether the effects observed in our study were specific to claustrum-PFC projections—and not a general consequence of activating any long-range glutamatergic input—we performed an additional control experiment targeting a distinct subcortical source: the dorsomedial thalamus (MD). Specifically, we expressed ChrimsonR in MD and stimulated thalamic axons in the dorsal prefrontal cortex (dPFC) using the same widefield optogenetic protocol and two-photon imaging approach used in our main experiments. Calcium activity was recorded from CaMKII+ neurons in dPFC during visual, optogenetic, and combined visual+opto conditions.

Our analysis of 436 cells across 3 mice revealed that, unlike claustrum stimulation, thalamic input did not lead to a significant increase in trial-to-trial variability or cross-cellular correlation (see **Supplementary Fig. 6**). Moreover, the distinct modulatory patterns observed with claustrum activation—such as opto-boosted responses and enhanced cross-population synchrony—were absent in the thalamus dataset. This result demonstrates that the modulatory effects we describe are not a general feature of all subcortical glutamatergic inputs to PFC but rather reflect a functionally specific role for the claustrum in shaping PFC sensory dynamics.

We have now included these findings in the revised manuscript and added the new data as **Supplementary Figure 6** as added in the Result section (Line 278-281)

Supplementary Figure 6. Optogenetic modulation of PFC responses during thalamus axon stimulation.

As an additional control experiment, we expressed ChrimsonR in the thalamus and delivered widefield optogenetic stimulation to thalamic axons in the dorsal prefrontal cortex (dPFC), while recording calcium activity from CaMKII+ cells in dPFC using two-photon imaging. A total of 436 PFC neurons were recorded across 3 mice. Panels A–C show heatmaps of trial-averaged $\Delta F/F$ responses from sensory-responsive neurons in response to visual stimuli alone (A), visual plus optogenetic stimulation (B), and the resulting modulation (C; Visual+Opto – Visual). Panels D–F show histograms of $\Delta F/F$ responses (D), absolute response magnitude (E), and coefficient of variation (F). Panel G presents violin plots of dynamic time warping (DTW) distance between visual and visual+opto conditions. Panels H–K and L–O show the same analysis for cells classified as opto-responsive (H–K) and opto-boosted (L–O), respectively. This dataset provides an additional comparison of how PFC neurons respond to visual input and optogenetic activation of thalamic inputs, using the same analysis framework as in Figure 3. This control dataset demonstrates that the effects observed in the main experiments are not generalizable across all subcortical inputs, and are not artifacts of the stimulation or imaging methodology, but rather reflect the specific modulatory role of claustrum input to PFC.

11) Consider double checking that the correlation in Figure 5F is indeed $r = -0.999$. This value seems biologically improbable given the spread of points.

We thank the reviewer for raising this point. The Pearson correlation between the visual response and opto-modulation is indeed highly negative ($r = -0.9999$), as confirmed by our direct calculation using trial-averaged responses. While the scatter plot may appear visually dispersed due to large N and axis scaling, the modulation value is computed as the difference between Visual + Opto and Visual responses, resulting in a strong negative dependency by design. Importantly, we used the exact same analysis code and pipeline for Figure 5F as we did for the subpopulation analyses shown in Figure 5D–E, where the correlation values are less extreme—highlighting that the observed difference arises from the population data itself rather than the method. The only difference across these analyses is the subset of animals and cells included. Interestingly, this negative dependency becomes even stronger in the silencing condition (OPN3 stimulation), possibly reflecting an even more direct subtractive effect on visually evoked activity.

12) In figure 3G, the cross-correlation matrix shows ~ 20k cells. This implies that the 20k cells were simultaneously imaged. Which was not the case as this is the number of cells across all fields of view. Therefore, I'm not sure how they can measure the pairwise correlation between cells that were not imaged simultaneously.

We thank the reviewer for this important clarification. The reviewer is absolutely correct that the full ~20,000 cells shown in Figure 3G were not recorded simultaneously. To address this concern, we revised our analysis to explicitly separate representational similarity across animals (RSA) from synchrony within simultaneously recorded fields of view (FOVs). We have now updated **Figures 3G–I** and the accompanying text to reflect this clarification. Result section (Line 283 - 309):

'Then, we assessed the collective impact of all responsive cells including the three identified subpopulations by examining the cross-correlation of trial-averaged responses across all neurons. To capture global changes in representational structure, we first computed a representational similarity analysis (RSA) across all recorded neurons from all animals (Fig. 3G). The matrix for visual stimuli showed distinct cell group clustering, suggesting a highly coordinated neural network (Fig. 3G, left). Conversely, the matrix for visual + opto stimuli demonstrated a more uniform distribution, with an absence of distinct clustering (Fig. 3G, right, using the same clustering as defined by the visual condition), indicative of a more homogenized representational landscape.

To further examine, we calculated within-FOV cross-correlation of trial-averaged responses for 1.5 second after the stimulation window to quantify local network synchrony. This analysis revealed that optogenetic stimulation increased the average pairwise correlation within each FOV (Fig.3I), consistent with a synchronization of activity across the recorded population. When we calculated the average change in correlation due to optostimulation across animals, both the global RSA (which summarizes representational similarity across all recorded neurons) and within-FOV cross-correlation showed a consistent increase for all responsive as well as non-responsive cells (Fig. 3H, significantly higher in all responsive cells than non-responsive cells, $p < 0.001$).

Notably, while the coefficient of variation analysis indicated higher variability in responses across trials within individual cells, the cross-correlation analysis, which examines trial-averaged responses, suggests increased synchrony among dPFC neurons during visual + opto trials. This indicates that the visual + opto stimuli facilitated a more homogenized and correlated interaction among all cells. This synergy among subpopulations likely contributes to refining dPFC activity, strengthening functional connections between neurons, and integrating them into a more cohesive network.'

13) The pie plots throughout are ineffective, confusing, and not proportional (Figure 1I). They detract from the paper.

Thank you for the suggestion. Figure 1.I is excluded from the figure.

14) The authors state that 14% of the cells (n = 5783), were classified as opto-boosted cells, and displayed responsiveness to visual stimuli only when preceded by photostimulation. I am confused by the term "preceded". Did they offset the opto stim from visual stimulation? There was no indication of this, and so when I read the paper I am confused.

Thank you for noticing this point. There was no time difference - this misleading wording is fixed. (Line 157)

15) What is the dynamic time warping analysis? This is central to the paper, yet the authors do not describe mathematically what this measure is or how it is calculated. No reference is given. Therefore, I cannot (and I did not) review anything related to this measure when I read the paper.

We thank the reviewer for this question. While the imaging sampling rate and stimulus alignment were tightly controlled in our experiments, dynamic time warping (DTW) was employed primarily to address trial-to-trial variability in the temporal dynamics of the neural responses rather than stimulus alignment errors. Even with precise stimulus triggering, calcium signals can exhibit temporal variability in rise time, peak latency, and decay dynamics across trials for a given cell. This variability may reflect meaningful differences in the underlying neural processes, such as variable recruitment of upstream circuits or fluctuations in internal state.

DTW allows us to compare the full temporal profile of calcium responses between trials, by flexibly aligning response trajectories to minimize temporal mismatches. By calculating the average DTW distance between all pairs of trials for each cell, we obtained a measure of how consistent the temporal patterning of responses was across repetitions, beyond simple amplitude variability. This provides complementary information to measures like coefficient of variation, which capture response magnitude fluctuations but are insensitive to timing variability.

In summary, DTW was used to quantify the temporal reproducibility of neural responses across trials, independent of minor latency shifts or shape variations, and not primarily to correct for stimulus-onset misalignment.

Method section updated accordingly (Line 749 - 759)

'To assess trial-to-trial variability in the temporal dynamics of neural responses, we employed dynamic time warping (DTW), a well-established method for aligning time series with temporal shifts or distortions (Berndt & Clifford, 1994; Müller, 2007). For each cell, fluorescence signals ($\Delta F/F$) were extracted for a 1-second window aligned to stimulus onset across all trials. DTW was then used to compute pairwise distances between the temporal response trajectories of all trial pairs for each cell. The DTW algorithm allows flexible nonlinear alignment of time series, accounting for small variations in response onset, peak latency, or shape across trials. For each cell, we quantified temporal variability as the average DTW distance across all trial pairs. This approach captures differences in temporal structure of calcium responses that may not be reflected by amplitude-based measures alone, providing a complementary index of response consistency across trials.'